# Structure-guided insights into heterocyclic ring-cleavage catalysis of the non-heme Fe (II) dioxygenase NicX

Gongquan Liu[1,3], Yi-Lei Zhao [1,3], Fangyuan He[2], Peng Zhang [2], Xingyu Ouyang[1], Hongzhi Tang [1✉] & Ping Xu [1]

Biodegradation of aromatic and heterocyclic compounds requires an oxidative ring cleavage enzymatic step. Extensive biochemical research has yielded mechanistic insights about catabolism of aromatic substrates; yet much less is known about the reaction mechanisms underlying the cleavage of heterocyclic compounds such as pyridine-ring-containing ones like 2,5-hydroxy-pyridine (DHP). 2,5-Dihydroxypyridine dioxygenase (NicX) from *Pseudomonas putida* KT2440 uses a mononuclear nonheme Fe(II) to catalyze the oxidative pyridine ring cleavage reaction by transforming DHP into *N*-formylmaleamic acid (NFM). Herein, we report a crystal structure for the resting form of NicX, as well as a complex structure wherein DHP and NFM are trapped in different subunits. The resting state structure displays an octahedral coordination for Fe(II) with two histidine residues (His[265] and His[318]), a serine residue (Ser[302]), a carboxylate ligand (Asp[320]), and two water molecules. DHP does not bind as a ligand to Fe(II), yet its interactions with Leu[104] and His[105] function to guide and stabilize the substrate to the appropriate position to initiate the reaction. Additionally, combined structural and computational analyses lend support to an apical dioxygen catalytic mechanism. Our study thus deepens understanding of non-heme Fe(II) dioxygenases.

[1] State Key Laboratory of Microbial Metabolism, Joint International Research Laboratory of Metabolic and Developmental Sciences, and School of Life Sciences and Biotechnology, Shanghai Jiao Tong University, Shanghai, People's Republic of China. [2] National Key Laboratory of Plant Molecular Genetics, CAS Center for Excellence in Molecular Plant Sciences, Institute of Plant Physiology and Ecology, Shanghai Institutes for Biological Sciences, Chinese Academy of Sciences, Shanghai, People's Republic of China. [3]These authors contributed equally: Gongquan Liu, Yi-Lei Zhao. ✉email: tanghongzhi@sjtu.edu.cn

Pyridine rings are primary components of pyridoxyl derivatives, natural plant alkaloids, and coenzymes. These compounds are more soluble in water, meaning they can spread into groundwater, which are hazardous to the health of humans and other organisms[1,2]. A pyridine ring opening reaction step is a common feature of most chemical and enzyme-based degradation processes for such pollutants, yet relatively little is known about such reactions, highlighting that gaining a clearer understanding of pyridine ring opening should enable development of management technologies.

2,5-Hydroxy-pyridine (DHP), a potential carcinogen, is a metabolic intermediate in the catabolism of many pyridine derivatives[3–7], which showed to cause DNA strand scission[8]. DHP is transformed to N-formylmaleamic acid (NFM) by a 2,5-DHP dioxygenase, an enzyme known as NicX from *Pseudomonas putida* KT2440 or Hpo from *P. putida* S16[5,9]. A previous biochemical study showed that this enzyme is a mononuclear non-heme iron oxygenase[9].

The superfamily of non-heme iron(II) enzymes catalyzes a wide range of oxidative transformations, ranging from the *cis*-dihydroxylation of arenes to the biosynthesis of antibiotics such as isopenicillin and fosfomycin[10–13]. These enzymes can be classified into several different groups based on their structural characteristics, reactivity, and specific requirements for catalysis, among them: (I) Extradiol cleaving catechol dioxygenases, (II) Rieske oxygenases, (III) Alpha-ketoglutarate dependent enzymes, (IV) Cysteine dioxygenases, and (V) Pterin-dependent hydroxylases[13,14].

A phylogenetic analysis of non-heme Fe(II) dioxygenases indicted that NicX is a member of a subclass of the non-heme iron dependent oxygenases (Fig. 1)[7]. Unlike other known non-heme Fe(II) enzymes, NicX catalyzes a pyridine ring-cleavage[7,15]. Notably, many ring-cleavage dioxygenases have been reported, including 2,3-HPCD from *Brevibacterium fuscum*[16], BphC from *Pseudomonas* sp. KKS102[17], PcpA from *Sphingobium chlorophenolicum*[18,19], and PnpCD from *Pseudomonas* sp. WBC-3[20]. The substrates for all of these enzymes contain aromatic rings. NicX has strong specificity for its DHP substrate; it cannot catalyze ring opening for pyridine-ring containing phenols, hydroquinones, or catechols[21]. The typical structural motif consists of a mononuclear iron(II) metal center that is coordinated by two histidine residues and one carboxylate ligand from either a glutamate or an aspartate residue. This structural motif has been coined the "2-His-1-carboxylate facial triad"[13]. In addition, a His/His/His triad coordinated Fe(II) has been found in cysteine dioxygenases and gentisate dioxygenase[22–24]. Interestingly, NicX Ser[302] coordinates the iron(II) ion; a similar metal ion-interacting serine residue has been reported for a dialkylglycine decarboxylase, Cu[+]-ATPases and for transcriptional activators like CueR and GolS[25–28]. Although NicX's DHP dioxygenase activity was enzymologically characterized in the 1970s, its structure has remained unsolved.

In this study, we present kinetics, mutational, and structural studies of NicX and clarify how NicX selectively recognizes DHP. We solved a resting homo-hexameric NicX structure as well as a NicX–DHP–NFM complex structure. We found that four residues of NicX (His[265], Ser[302], His[318], Asp[320]) coordinate the iron (II) ion. We also found that Leu[104] and His[105] adapt different conformations in DHP-bound monomers vs. NFM-bound monomers. In addition, molecular modeling, molecular dynamics simulations and quantum mechanics calculations were combined with the crystallographic 3D structural data to propose the

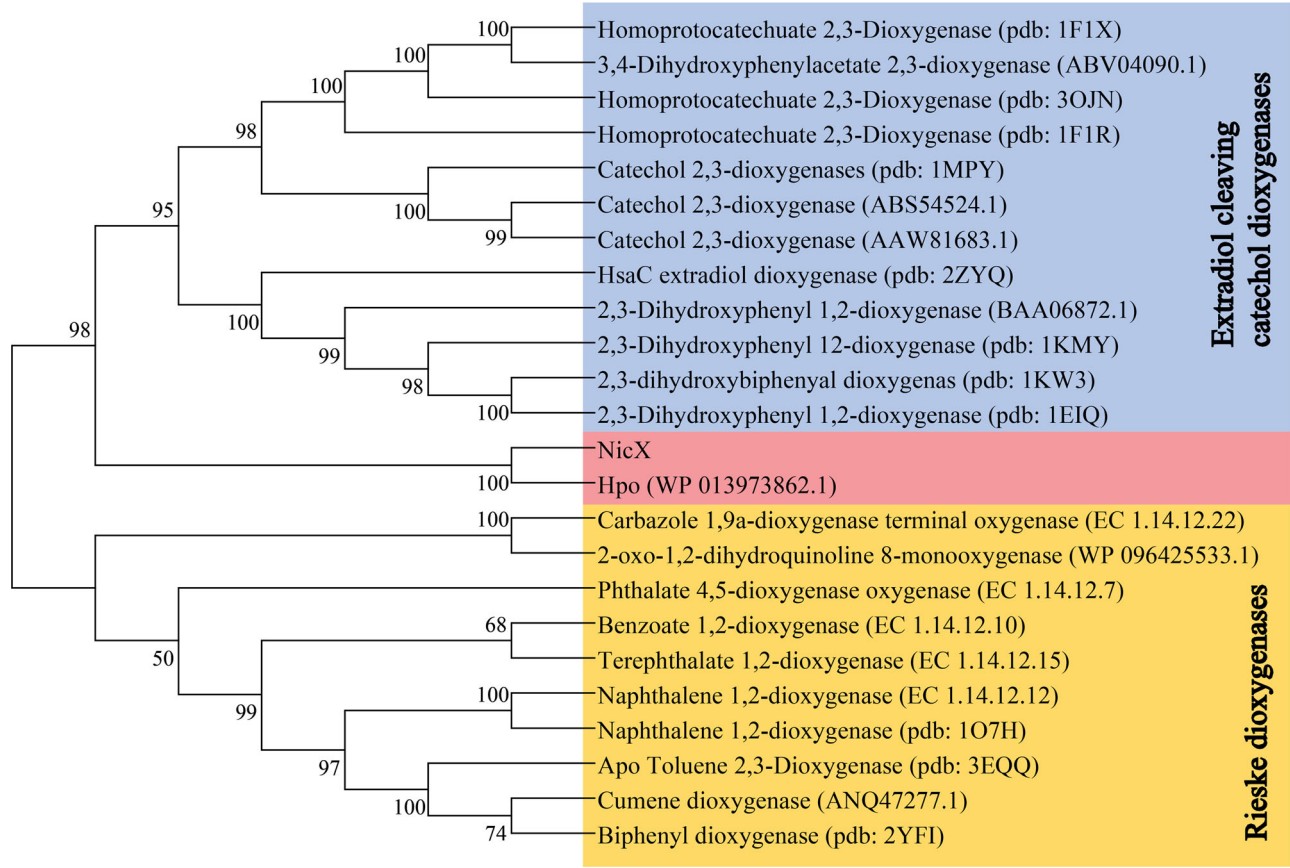

**Fig. 1 Phylogenetic analysis of NicX.** Phylogenetic tree of NicX with selected dioxygenases constructed by using neighbor-joining method. GenBank accession numbers or pdb numbers are shown at the end of each name. Bar represents 1.0 amino acid substitutions per site.

**Table 1 Data collection and refinement statistics.**

| | SeMet-NicX | NicX | NicX in complex with DHP and NFM |
|---|---|---|---|
| ***Data collection*** | | | |
| Space group | $P2_12_12$ | $P2_12_12$ | $P2_12_12$ |
| Wavelength (Å) | 0.97918 | 0.97892 | 0.97918 |
| Cell dimensions | | | |
| a, b, c (Å) | 125.92, 144.13, 118.89 | 125.961, 143.75, 118.69 | 126.67, 145.51, 118.98 |
| α, β, γ (°) | 90.00, 90.00, 90.00 | 90.00, 90.00, 90.00 | 90.00, 90.00, 90.00 |
| Number of molecules/asymmetric unit | 6 | 6 | 6 |
| Resolution range (Å) (outer shell) | 50-2.28 (2.40-2.28) | 50.00-2.00 (2.03-2.00) | 50-2.20 (2.24-2.20) |
| Completeness (%) (outer shell) | 99.2 (99.6) | 100.0 (97.3) | 99.8 (99.2) |
| Redundancy (outer shell) | 6.0 (6.2) | 12.3 (12.3) | 12.8 (13.1) |
| Total observations | 585,113 | 1,763,979 | 1,453,302 |
| Unique reflections | 98,134 | 142,933 | 113,279 |
| Wilson B factor (Å$^2$) | 50.13 | 22.91 | 27.51 |
| $R_{merge}$ (%) (outer shell) | 9.1 (65.1) | 10.1 (95.3) | 10.6 (95.9) |
| $I/\sigma_I$ (outer shell) | 8.7 (3.0) | 25 (2.67) | 24.4 (3.7) |
| ***Refinement*** | | | |
| Resolution range (Å) | 30-2.28 | 47.41-2.00 | 27.59-2.20 |
| $R_{work}/R_{free}$ (%) | 21.1/26.7 | 17.12/21.33 | 18.3/23.8 |
| Average B-factors(Å$^2$) | | | |
| Protein, metal ion, water, substrate/product | 67.3, 97.2, 55.1, - | 27.75, 38.04, 40.28, - | 29.52, 45.6, 38.8, 42.9/55.1 |
| Root mean square deviations | | | |
| Bond angles (°), Bond lengths (Å) | 0.014, 1.644 | 0.010, 1.625 | 0.008, 0.908 |
| Number of atoms | | | |
| Protein/substrate/water | 16,371/0/241 | 16,334/0/1514 | 16,576/52/940 |
| Ramachandran plot | | | |
| Most favored, allowed, disallowed (%) | 95.1, 4.7, 0.1 | 95.81, 3.66, 0.53 | 95.38, 4.23, 0.38 |

plausible catalytic mechanisms of NicX. Our study of NicX deepens understanding of non-heme Fe(II) dioxygenases.

## Results

**Structural determination and overall structure**. Seeking to better understand how NicX selectively recognizes its substrate DHP, we adopted a selenomethionine (SeMet) phasing strategy in which we initially determined the crystal structure of a SeMet-substituted resting NicX to a resolution of 2.28 Å using the single-wavelength anomalous dispersion (SAD) method (Table 1). Subsequently, the crystal structures of NicX and complex NicX–DHP–NFM were solved using coordinates of SeMet-NicX (Table 1). There are six molecules in an asymmetric unit in all the structures, consistent with gel filtration chromatography results revealing that NicX occurs as a homohexamer in solution (Fig. 2a)[29].

The N-terminus of each NicX subunit has a 150-residue domain comprising 12 α helices and 19 β-strands (residues 2–151): this domain mainly consists of continuous αβ motifs. Each subunit also has a 199-residue C-terminal domain (residues 152–350), which contains two antiparallel β-sheets (β5-β6-β10-β13-β14-β18-β19/β7-β9-β11-β12-β15-β16-β17) (Fig. 2b).

**Structure of the NicX–DHP–NFM complex**. We examined the aforementioned NicX–DHP–NFM complex structure and found that only four subunits (C, D, E, and F) of the six constituent molecules in this structure bind with substrate DHP; the other subunits (A and B) contain the product NFM. This phenomenon of heterogeneous monomer binding patterns for a multi-homomeric enzyme is not surprising, with similar reports for other metal iron-dependent dioxygenases[30–33]. In the NicX–DHP–NFM complex, each subunit contains a fully occupied Fe(II), each of which coordinates with six ligands: four residues and two waters (Fig. 2c). Notably, site where water 2 is positioned exhibits elongated density in our models for one of the DHP-bound subunits (subunit D). This density was modeled as a

water molecule, because the density for the possible oxygen is not clear while the previously reported trend that non-heme Fe(II) enzymes typically requires the presence of a substrate for oxygen binding to occur[34,35].

Strikingly, the arrangement of the 14 Å-deep pockets in the NicX surface (Fig. 2d) changes based on the binding activity of a given subunit. That is, NFM-bound subunits appear the same as the resting state: there are apparently two channels at the enzyme surface near the active site, which are separated by residues His[105] and Glu[308], in close proximity to Leu[104] ("open" state; Fig. 2e). However, only one of two channels occurs in the DHP-bound subunits (henceforth "channel I", "close" state; Fig. 2f).

**Characteristics of the Fe (II) coordination in NicX**. There is a conspicuous, deep pocket (~14 Å in depth) on the surface of NicX that harbors its catalytic active site (Fig. 2d). Here, a ferrous ion is held in place via coordination involving four residues (His[265], Ser[302], His[318],and Asp[320]) and two waters. Interestingly, the coordinating ligand Ser[302] has not been previously reported in studies of other ferrous ion-dependent dioxygenases (Fig. 2c). To verify that these four residues directly participate in the iron coordination (rather than crystal packing), we conducted alanine screening mutation analysis. We observed a complete loss of activity for the H265A, S302A, H318A, and D320A mutants in enzymatic assays (Supplementary Table 1). Moreover, ICP-MS analysis revealed iron signals for the wild type enzyme but no such signals for any of these four mutant variants, and the circular dichroism analysis showed that the secondary structures of the mutation variants were not changed (Supplementary Table 2, Supplementary Table 3, and Supplementary Fig. 1). Collectively, these results verify the direct participation of these four residues in iron coordination with the C-terminal domain of NicX. It should be noted that although crystals were soaked in the buffer contain Fe$^{2+}$, we were not able to experimentally determine the oxidation state of the iron in the crystal structures.

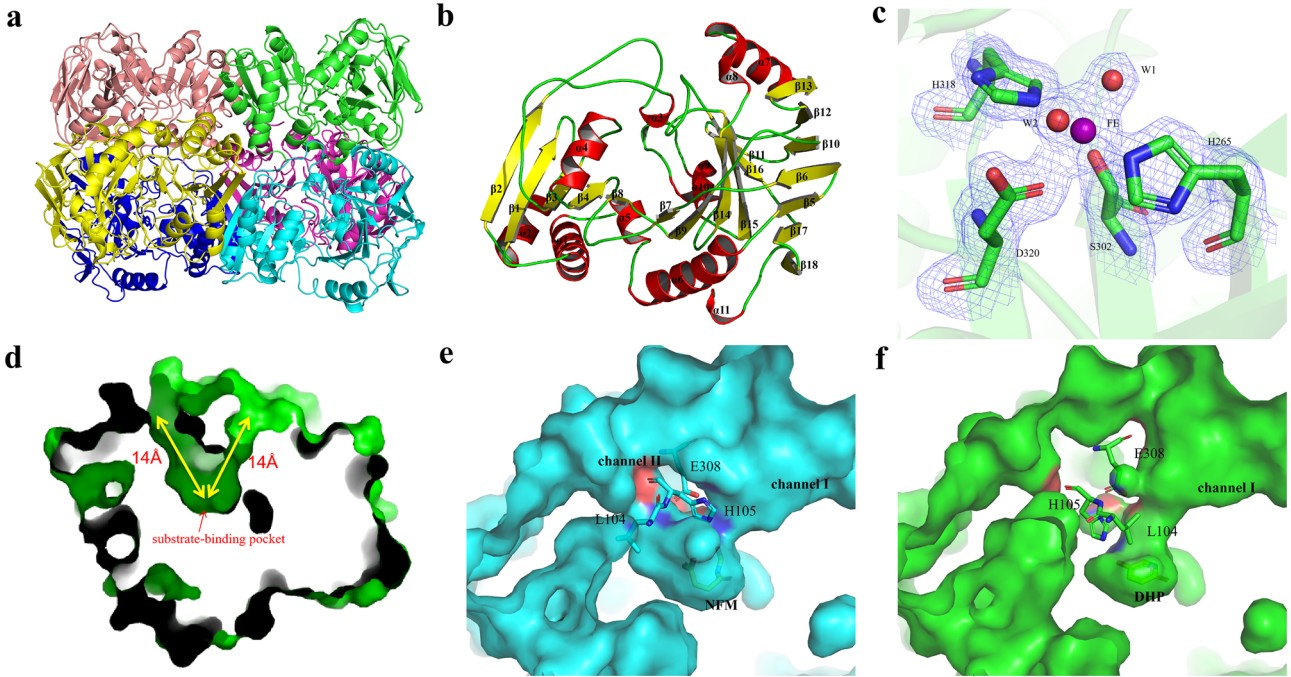

**Fig. 2 Structure of NicX. a** Overall structure of NicX. Ribbon plot representation of the NicX hexamer. **b** Overall structure of a protomer of NicX. The α-helices, β-sheets, and loops are in red, yellow, and green, respectively. Secondary structure elements of NicX are labeled. **c** Coordination sites of NicX-Fe (II). The $2F_O$-$F_C$ electron density map is contoured at 1σ, colored in blue; water 1 molecule is opposite Asp[320], while water 2 molecule is opposite Ser[302]; green, red, blue, and purple represent carbon, oxygen, nitrogen, and Fe atoms, respectively. **d** The 14 Å-deep substrate-binding pocket on the NicX surface, two channels are observed. **e** Two channels are separated by residues His[105] and Glu[308], observed in resting or NFM complex structure. **f** In the DHP-bound subunits, channel II was blocked by the Leu[104] and His[105] residues.

**Substrate and product binding in the complex structure.** In DHP-bound subunits of the complex, two hydrogen bonds are formed between Glu[177] and the N1 and the O1 of DHP (2.5 and 3.0 Å), one hydrogen bond is formed between His[189] and O1 of DHP (2.6 Å) (Fig. 3a). Thus, these hydrogen bond interactions may have an indispensable effect on the proper pre-catalytic positioning of the substrate. We explored the functional relevance of Glu[177] and His[189] with alanine mutant variants of these residues (as well as a NicX[E177A&H189A] mutant) which we tested with in vitro enzymatic assays with purified enzymes. Wild type NicX was purified to more than 95% homogeneity after recombinant expression in *E. coli* cells, and exhibited apparent $K_m$ and $V_{max}$ values for conversion of DHP to NFM of 94.9 ± 3.84 μM and 58.62 ± 0.95 U·mg$^{-1}$ (Supplementary Table 1), respectively. The purified NicX[E177A&H189A] variant completely lost its catalytic activity for DHP. The NicX[E177A] and NicX[H189A] variants enormously reduced activities (reduced to only 2.7 and 6.9% of the wild type), and exhibited 2.7-fold and 2.2-fold higher $K_m$ values compared with the wild type, respectively.

It is intriguing that a NicX[R293A] variant completely lost activity for DHP (Supplementary Table 1). Observation in structure that the side chain of Arg[293] is positioned about 4.3 Å distant from the center of DHP pyridine ring, and this residue does not apparently change its position or orientation in the resting or NFM-bound subunits (Fig. 3a). Moreover, Arg[293] forms two salt bridge interactions with the side chain carboxyl group of the confirmed Fe-binding residue Asp[320] (2.7 and 3.5 Å) (Fig. 3a). Pursuing this, we conducted ICP-MS assays and found that the NicX[R293A] variant lost the ability to bind a ferrous ion (Supplementary Table 2). This result indicates that the role of Arg[293] may be primarily steric, apparently functioning to position the Asp[320] correctly for ligation.

Analysis of the ring-open product NFM in the active sites of the A and B subunits of the complex structure revealed that the pyridine ring of DHP has been cleaved between the C5[DHP] and C6[DHP] carbons to form the product NFM (Fig. 3b). The carboxide derived from the C2[DHP] carbon of the product interacts particularly strongly with residues Glu[177] (2.8 Å) and His[189] (2.9 Å) (Fig. 3b), so that the product stably binds to the enzyme, additionally supported by NFM's *cis*-carbon double bond[7]. As mentioned above, Fig. 3b provides direct evidence of ring fissure.

**A conformational change of Leu[104] and His[105].** A careful examination of structure shows that there is a conformational change for the Leu[104] and His[105] in the substrate-bound vs. both the resting structures and the product-bound subunits (Fig. 3c, Supplementary Movie 1). It is a surprise to find that a hydrophobic path that goes straight to the active center of ferrous ion when channel II is closed in the E subunit (Supplementary Fig. 2). However, this hydrophobic path is blocked by Leu[104] in resting state or product NFM binding state, suggesting that a hydrophobic path appears induced by the conformational change of Leu[104]–His[105] (Supplementary Fig. 2). Leu[104] seems to interact with DHP through the side chain pyridine ring (3.9 Å) via both CH-π[36] and van der Waals interactions (Fig. 3a). His[105] forms a hydrogen bond with the substrate DHP (3.1 Å) (Fig. 3a). We performed in vitro assays with mutant variants to confirm functional contributions from these residues in NicX's enzymatic activity. The NicX[H105A] variant completely lost enzyme activity, and the NicX[L104A] variant showed a dramatically decreased activity (down to only 3% of the wild type) and exhibited a 1.2-fold higher $K_m$ value compared with the wild type. Notably, the NicX[E308A] variant lost about half of its activity, and had a $K_m$ value very similar to wild type NicX, results suggesting that perhaps Glu[308] does not participate directly in the recognition of DHP, but rather affects enzymatic activity via an interaction with His[105].

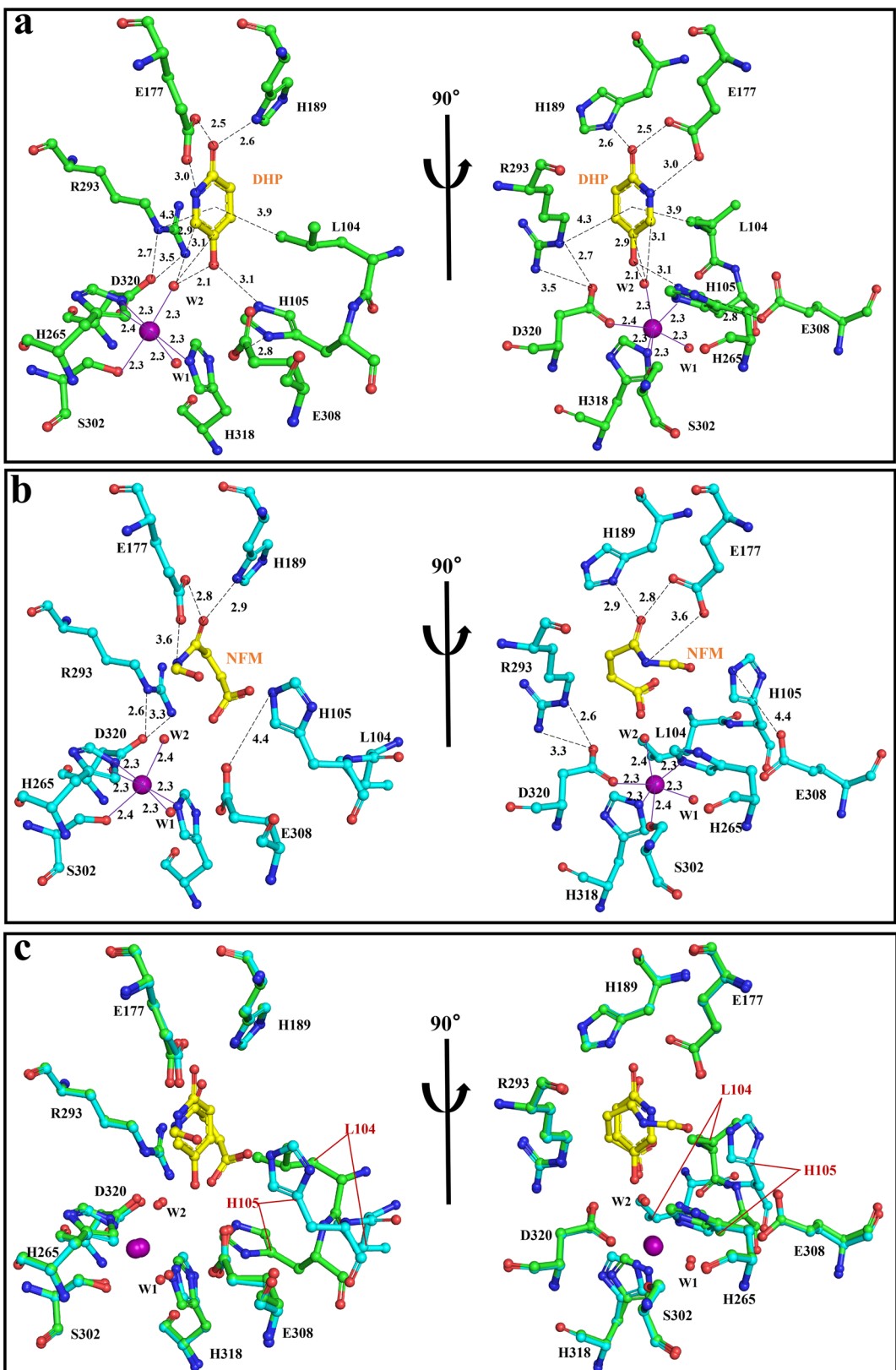

**Fig. 3 Active site of NicX. a** Active site of NicX with bound DHP. The carbon atoms of DHP are in yellow. Green, red, blue, and purple represent carbon, oxygen, nitrogen, and Fe atoms, respectively. **b** Active site of NicX with bound NFM. The carbon atoms of NFM are in yellow. Cyan, red, blue, and purple represent carbon, oxygen, nitrogen, and Fe atoms, respectively. **c** A stereoview of the DHP bound subunit (green) and NFM bound subunit (cyan), the superimposition was done on the whole subunit. Iron and solvent molecules are shown as purple and red spheres, respectively. DHP/NFM interacts with residues were indicated as black dotted line, the coordination bonds of ferrous ion are shown as solid lines in purple, distances are given in angstroms.

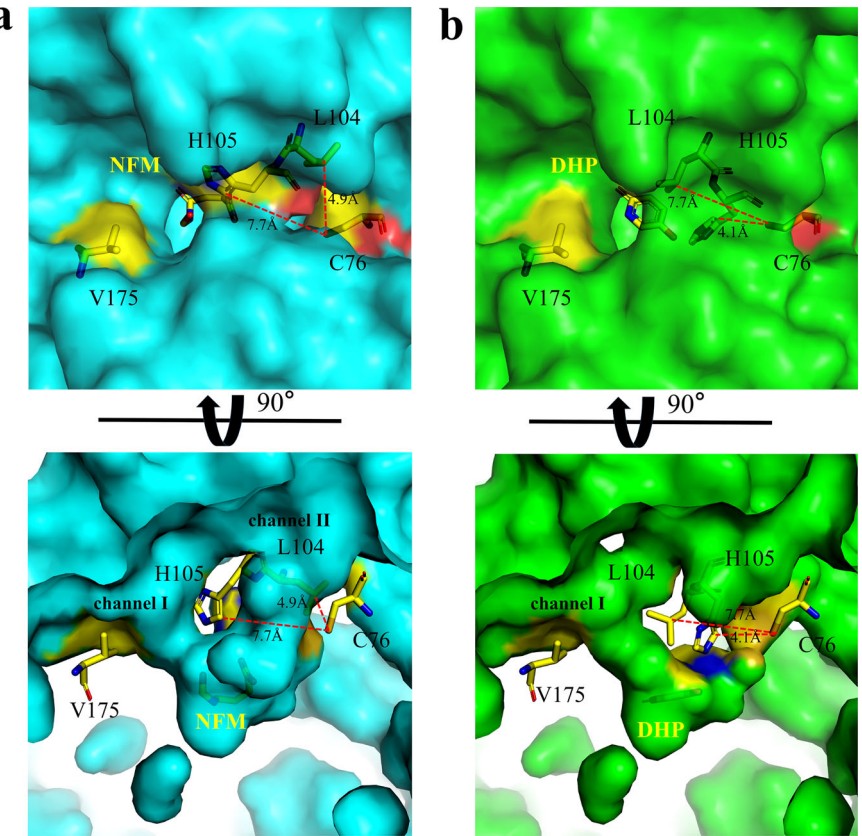

**Fig. 4 A conformational change of L104 and H105. a** In NFM bound subunits, Cys[76] is close to Leu[104] at a distance of 4.9 Å, away from His[105] at a distance of 7.7 Å. **b** In DHP bound subunits, Cys[76] is close to His[105] at a distance of 4.1 Å, away from Leu[104] at a distance of 7.7 Å. Yellow, red, blue, and orange represent carbon, oxygen, nitrogen, and sulfur atoms, respectively.

Based on structural analysis, it could be seen from the NFM bound subunits that Cys[76] is close to Leu[104] at a distance of 4.9 Å, away from His[105] at a distance of 7.7 Å (Fig. 4a, b). In contrast, in DHP bound subunits, Cys[76] is close to His[105] at a distance of 4.1 Å, away from Leu[104] at a distance of 7.7 Å. This means that the conformational changes of Leu[104]–His[105] may be influenced by Cys[76] (Fig. 4a, b). It seems that the enzyme activity could also be influenced by Val[175] in channel I but far away from active sites. To verify the above idea, Val[175] in channel I and Cys[76] in channel II were selected for mutation (Fig. 4a, b). The NicX[V175F] variant activity reduced to only 21.8% of the wild type, but $K_m$ values did not change much compared with the wild type, respectively (Supplementary Table 1). In contrast, the point mutations of C76Q or C76E completely abolished the enzymatic activity of NicX; however, it was notable that the NicX[C76A] variant retained the capability of transforming DHP into NFM (Supplementary Table 1). From the result of the mutation experiments, it appears that channel II has a more significant effect on the enzyme activity. Thus, we speculate that the bulky residues (glutamine or glutamic acid) take up more space than cysteine, and affect the range of movement of Leu[104] and His[105], causing it to be unable to guide and stabilize the substrate to the appropriate position to initiate the reaction.

**Structural and computational studies for the NicX mechanism.** Although NicX is the subtype defining member for non-heme iron(II) ring-cleavage dioxygenases, its overall catalytic mechanism bears some similarities with other non-heme iron(II) dioxygenases. Based on the studies of other Fe(II)-dependent dioxygenases[12,13,30–32,37], and the computational analysis on our structures, we proposed two possible equatorial and apical dioxygen catalytic mechanisms, and calculated energetics of these pathways (Fig. 5, Supplementary Fig. 3).

If the dioxygen takes the opposite position of Asp[320] ("equatorial position"), there are two possibilities for substrates to coordinate with metal ions: the hydroxyl group is at position 5 to chelate with Fe (II) (Pathway IA), or the N atom on the pyridine ring can coordinate with Fe (II) (Pathway IB). Subsequently, two one-electron transfers are needed to form the peroxide intermediate, which can be followed by a Criegee rearrangement to yield a 7-membered-ring lactone and a ring-opened product; either of these scenarios would result in a substrate-bound iron arrangement similar to the classic extradiol catechol dioxygenases (Pathway IA & IB). Such a position would suggest a *trans*-effect for Asp[320] in promoting the subsequent O-O cleavage reaction. However, this situation would require the DHP substrate to drop from the hydrogen-bonding network comprising His[105]–Glu[177]–His[189], which would cause considerable destabilization of the reacting substrate–enzyme complex (Pathway IA & IB) (Supplementary Fig. 4a, b). Alternatively, the dioxygen molecule could be positioned opposite to Ser[302] ("apical position"), in which case the C5 and C6 atoms of DHP would be adjacent to the dioxygen molecule. The apical arrangement would be similar to a P450-like arrangement, and Ser[302] could exert a cystine-like catalytic role (Pathway II) (Supplementary Fig. 4c). More interestingly, the pyridinyl N-H could also plausibly involve in the proton transfer to activate the O-O cleavage in Pathway II, equivalent to imidazolyl N-H of the histidine residue in extradiol dioxygenases. Further investigation on the roles of Arg[293]–Asp[320]–DHP[N-H] is needed to illustrate the details of

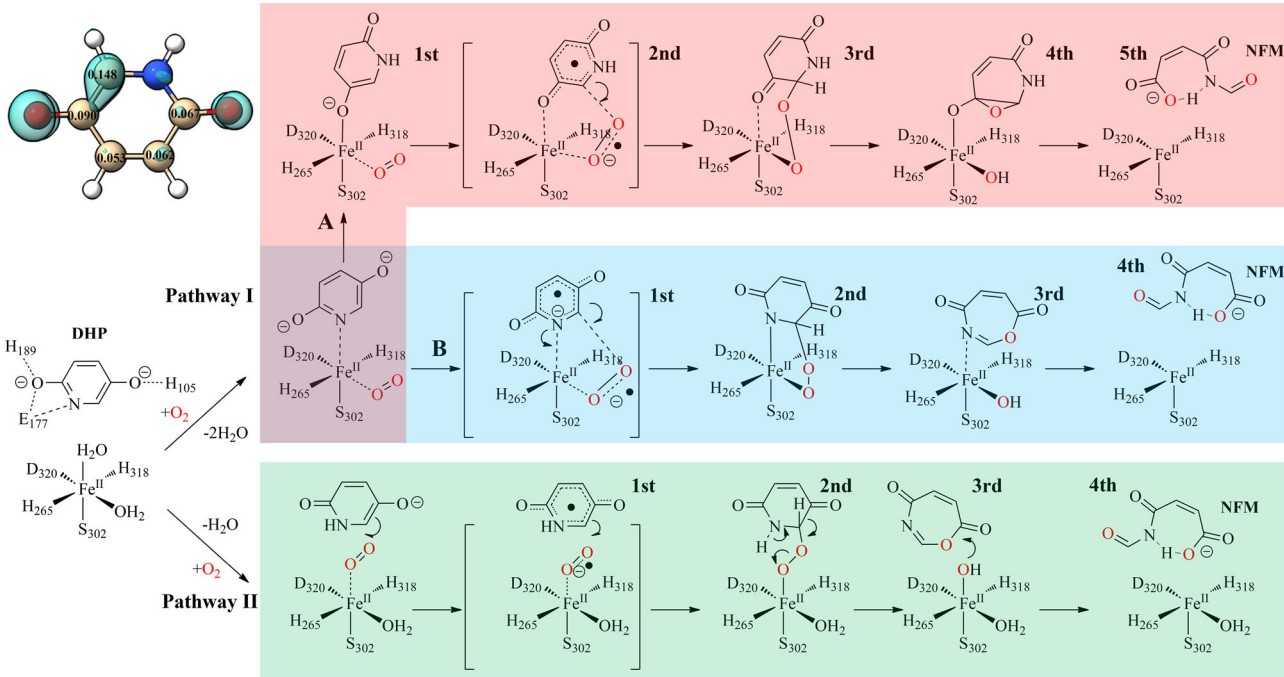

**Fig. 5 Plausible pathways for NicX-catalyzed DHP degradation, in which dioxygen attacks either from the equatorial position (Pathway IA, colored in pink; Pathway IB, colored in cyan), or the apical position (Pathway II, colored in light green).** Pathway IA and IB denote that the nitrogen and oxygen atoms of DHP bind the metal ferrous center, respectively. In Pathway II, the substrate DHP does not directly chelate with the ferrous ion. The possibly one-electron transfer radical species are drawn in square brackets. The C6 is the most vulnerable position in DHP, with an $f^-$ value of 0.148 based on Fukui function analysis in the upper left corner (C2, 0.067; C3, 0.062; C4, 0.052; C5, 0.090; C6, 0.148).

proton transfer for the O-O cleavage in the ring-open process. In a preliminary study of the conversion of the quintet peroxide intermediate to a 7-membered-ring lactone, both concerted, and stepwise mechanisms were examined (Pathway II, Supplementary Fig. 5). These calculations indicated a preference for a stepwise O-O cleavage; and low-lying septet state might promote spin-flip during the breaking and formation of multiple bonds (Supplementary Fig. 5).

## Discussion

Multiple crystal structures of dioxygenases that can catalyze the ring-opening cleavage of aromatic compounds have been reported[30–32], but we are unaware of any reported crystal structures for an enzyme capable of catalyzing cleavage of a pyridine ring. It is notable that a phylogenetic analysis of non-heme Fe(II) dioxygenases indicated that NicX and Hpo comprise a subclass of non-heme iron dependent oxygenases (Fig. 1). Despite extensive efforts, we were unable to obtain a structure for Hpo, and structural comparison using the DALI server failed to identify any obvious homologs of full-length NicX.

Another notable observation from our study that a conformational change in the Leu[104] and His[105] residues induce a major change in channel II; this change creates a hydrophobic path that goes straight to the active ferrous ion. Based on in vitro assays, we confirmed that His[105] has a dramatic effect on enzyme activity. Previous studies have proposed that His residues could facilitate alkylperoxo intermediate breakdown, potentially via protonation to form a gem-diol intermediate[38]. We therefore examined whether His[105] exerts this type of effect in NicX's mechanism of action. However, we found that when His[105] was mutated into aprotic amino acids (e.g., our NicX[H105M] and NicX[H105F]), these variants retained their catalytic activity for producing NFM from DHP (Supplementary Table 1). We interpret these results to rule out the participation of His[105] in proton

transfer during NicX catalysis. Our analyzes support speculation about two possible functions of the conformational change for Leu[104]–His[105]. The first is the aforementioned creation of a hydrophobic path that could greatly facility the direct delivery of a dioxygen molecule to the iron center (Supplementary Fig. 2). The second potential function of the conformational change could be to guide and assist to bind the DHP substrate molecule to a pre-reaction position, besides Glu[177]–His[189]; this may facilitate activation of the dioxygen and promote reaction initiation. Our site-directed mutagenesis results show that NicX[C76Q] and NicX[C76E] variants are inactivated provides further (albeit indirect) evidence of the contribution of Leu[104] and His[105] in orienting the substrate DHP.

It is also fascinating to find that the NicX ferrous center coordinates with only four residues (His[265], Ser[302], His[318], and Asp[320]). Ser has been reported as a ligand for proteins including dialkylglycine decarboxylase, Cu[+]-ATPases, and transcriptional activators[25–28]; however, it has not been reported in non-heme dioxygenases, suggesting that NicX has a previously unknown fold architecture and active site environment for non-heme iron (II) dioxygenases. We examined site-directed mutant variants of NicX and confirmed the indispensability of these four residues for Fe(II) coordinating by using enzymatic assays, ICP-MS assays, and circular dichroism analyzes.

It is intriguing that NicX has not been reported to have activities for substrates other than DHP. Our study indicates that this substrate specificity is strongly impacted by NicX's substrate-binding sites, including His[105], His[189], and Glu[177], each of which is involved in the unique hydrogen bonding network. The O1 atom of DHP interacts with the residues of Glu[177] and His[189], and the N atom of DHP affects the electron distribution over the pyridine ring. Given that N atoms are more electronegative than carbon atoms, these factors apparently contribute to controlling the ionic configuration of the substrate DHP; thus the N atom of DHP is placed adjacent to

the Glu[177] side chain carboxyl group, rather than some other orientation. We consider the different catalytic pathways of NicX, wherein the dioxygen occupies the equatorial position (Pathway I) or apical position (Pathway II); these pathways are ultimately downhill energetically and cannot be distinguished based on energetics alone. However, the specific substrate orientation in the co-crystal structure motivated us to reconsider the possibility of Pathway II. First of all, the arrangement in Pathway II would allow maintenance of the specific hydrogen bonding network for holding the DHP substrate in place; in contrast, the DHP molecule would have to undergo a "big flip" and shift downwards by ~4 Å to bind the metal and contact the equatorial dioxygen in Pathway I. Importantly, this would require disturbing the specific hydrogen bond network observed in the crystal structure. Alternatively, in Pathway II the bridging peroxo can undergo a ~0.5 Å shift of the substrate and a ~20° rotation of the substrate, a scenario that would retain the intermediate in the hydrogen bond network. Second, the first one-electron transfer of the DHP substrate seems to occur at the moiety of O-C = N based on the pKa ($pK_{a1}$ = 8.56) and Fukui function ($f$ = 0.148) (Fig. 5), and the dioxygen molecule could occupy the vacancy between C6---Fe and mediate the ignition of the DHP-$O_2$-Fe triad. Third, the consequent O-O cleavage could be promoted by neighboring proton donor candidates such as imidazolium of His[105], neutral form of Asp[320], aromatic N-H of DHP, and even guanidinium of Arg[293] [39,40]. Moreover, a crystal structure-based CAVER analysis indicated that the dioxygen tunnels terminate at a position opposite to Ser[302] rather than Asp[320] (Supplementary Fig. 2). Thus, both structural analysis and preliminary computation lend support the apical hypothesis, but additional theoretical calculations will be required for further validation on these possible mechanisms.

In past studies, the mechanisms of homogentisate 1,2-dioxygenase (HGDO, PDB code 4AQ6 [https://doi.org/10.2210/pdb4AQ6/pdb]) and homoprotocatechuate 2,3-dioxygenase (HPCD, PDB code 2IGA [https://doi.org/10.2210/pdb5XRN/pdb]) have been investigated by trapping different reaction cycle intermediates in different subunits of a single crystal[30,32]. HPCD is a prototypical type I extradiol dioxygenase acting on a catechol-type substrate, while HGDO reacts with a benzene-type substrate containing hydroxyl group in the *para* position. Both enzymes share things in common: (i) both catalyze the oxidative aromatic ring cleavage; (ii) both have a mononuclear iron(II) metal center that is coordinated by two histidine residues and one carboxylate ligand; (iii) their substrates act as dentate ligands when bound to the active site Fe (II) ion; (iv) their mechanisms both feature an attack of a superoxo ligand on their substrates at a hydroxylated carbon. It should be emphasized that both our crystal structure data and computational studies highlight differences in the apparent reaction mechanism of NicX (Fig. 5) compared to HPCD and HGDO. Specifically, (i) NicX is able to catalytically crack a pyridine ring substrate; (ii) NicX has a mononuclear iron(II) metal center that is coordinated by two histidine residues, one carboxylate and a serine residue; (iii) DHP does not directly chelate ferrous ion; (iv) the reaction between superoxide and DHP proceeds by reaction at the C6 atom of DHP, not the OH-group carrying C5 atom. In light of these differences, it is not surprising that our crystal structures and computational studies indicate clear distinctions for the proposed reaction mechanism of NicX vs. the reaction mechanisms of HPCD and HGDO.

In summary, we here determined the crystal structure of a pyridine ring-cleavage enzyme. Our structural and computational studies of NicX from *P. putida* shed lights on the ring-cleavage mechanisms used by dioxygenases; our study deepens understanding about how non-home Fe(II) ring-cleavage dioxygenase family enzymes interact with their aromatic and heterocyclic substrates.

## Methods

**Chemical reagents.** 2,5-Dihydroxypyridine was purchased from Aladdin. SeMet was purchased from Acros Organics. Crystallization screens were obtained from Hampton Research. All other chemicals were obtained commercially.

**Plasmid preparation, recombinant expression, and protein purification.** The DNA fragment encoding full-length WT *P. putida* KT2440 NicX was cloned into the pET-24a (Novagen) vector between the *Nde*I *Hind*III sites using DNA primers 5′-agtcatatgccggtgagcaatgcacaa-3′ and 5′-tataagctttcgcgctcgcgactcct-3′ (bearing a sequence encoding 6 C-terminal His-tags) (Supplementary Table 4). All mutants were generated using a whole-plasmid PCR and *Dpn*I digestion method, and the sequences of the constructs were verified via Sanger sequencing, primer pairs used for installing each mutation were shown in Supplementary Table 4. These plasmids were transformed into *E. coli* BL21(DE3) cells. Cells possessing plasmids for the wild type and the mutant variant of NicX were grown at 37 °C to an $OD_{600}$ of 0.8, after which they were subjected to overnight induction at 16 °C with 0.4 mM isopropyl β-D-1-thiogalactopyranoside. Centrifugation for 15 minutes at 4,000×g was used to harvest the cells, with the resulting cell pellets resuspended in a binding buffer (25 mM Tris-HCl, pH 8.0, 300 mM NaCl, and 20 mM imidazole, 0.5 mM PMSF, 1 mM $FeCl_2$, 10 mM β-Mercaptoethanol) and subsequently lysed using a cell homogenizer. We then centrifuged the cell lysate and purified the supernatant with $Ni^{2+}$-NTA affinity chromatography (Qiagen) and Superdex200 gel filtration chromatography (GE healthcare). The gel filtration buffer was comprised of 200 mM NaCl, 20 mM Tris-HCl (pH 8.0), and 2 mM dithiothreitol (DTT). Subsequently, fractions with bands putative recombinant NicX proteins (~39 kDA, assessed via SDS-PAGE) were combined and concentrated to 25 mg ml⁻¹, and were then flash-frozen in liquid nitrogen and stored at −80 °C.

Note that a SeMet-substituted NicX variant was expressed using the methionine-autotrophic *E. coli* strain B834(DE3) cultured in M9 minimal media, and was purified using the same procedure as for the native protein, except that 5 mM DTT was used in the Superdex200 gel filtration buffer. Purity was monitored for all protein preparations based on SDS-PAGE, and protein concentrations were determined using a NanoDrop2000 spectrophotometer.

**Enzyme assay.** Purified WT and NicX mutant variant proteins used in the in vitro enzymatic activity assay. Activity assay was performed at 25 °C, monitoring the absorbance at 320 nm using a UV-Vis 2550 spectrophotometer[21]. The reaction mixture contained 20 mM Tris-HCl (pH 8.0), 50 μM $FeCl_2$, as well as the DHP substrate and the enzyme (concentrations of both depending on the nature of the particular assay) in a total volume of 800 μl. The enzyme was incubated with $FeCl_2$ (1 mM) for 1 min, and the reaction was initiated via the addition of DHP. Activity was defined as the amount of enzyme that catalyzes the conversion of 1.0 μmol of DHP in 1 min. The assay was performed independently three times, and data are presented as means ± the standard deviations.

**Crystallization, data collection, and structure determination.** Crystals of NicX were grown from a 1:1(v/v) mixture of a NicX protein solution (25 mg ml⁻¹), a reservoir solution (0.2 M Succinic acid (pH 7.0) and 20% (w/v) PEG 3350), using the hanging-drop vapor-diffusion method at 20 °C. The SeMet-NicX variant was crystallized in the reservoir solution containing 0.2 M Sodium formate (pH 7.0) and 20% (w/v) PEG 3350. The complexes were prepared by soaking NicX crystals in cryoprotectant buffer supplemented with 5 mM $Fe^{2+}$ for about 5 min prior to soaking crystals in cryoprotectant buffer supplemented with 5 mM DHP and 20 mM sodium dithionite. After soaking for 30–60 min, crystals were rapidly transferred into mother liquor solutions containing 25% glycerol prior to cryocooling in liquid nitrogen.

Crystal diffraction data sets of the native and SeMet-NicX were collected at the BL17U1 and BL19U1 beamlines of the Shanghai Synchrotron Radiation Facility by using an ADSC Quantum 315r detector or a DECTRIS PILATUS3 6 M detector at a wavelength of 0.97918 Å at 100 K. All data were processed and scaled using the HKL3000 program[41]. The SAD phases were determined using the Autosol module of PHENIX[42]. After the model-building with Coot[43] and refinement with REFMAC[44]. The structures of NicX and complex NicX–DHP–NFM were solved using coordinates of SeMet-NicX; the substrate/product molecules were placed in the model based on the Fourier difference map, and refined using the geometric restraints prepared using REEL in Phenix[45]. The figures were prepared using PyMOL. Crystallographic statistics are listed in Table 1. The resulting coordinates and structure factors have been deposited in the Protein Data Bank (Protein Data Bank codes: 7CNT、7CUP、7CN3).

**Inductively coupled plasma mass spectrometry.** An inductively coupled plasma mass spectrometer, iCAP Qc (Thermo Fisher scientific, USA), with KED (Kinetic energy discrimination) system was used for detection. Before the ICP-MS assays, the purified mutant protein was incubated with an appropriate amount of ferrous ion for at least 30 min (the molar ratio of protein to ferrous ion was 1:5) and again

subjected to a Superdex 200 column. 200 µl of protein sample was added 0.5 ml HNO₃ and let it sit for half an hour prior to heat the water bath for 90 min. Before use, sample volume was diluted to 5 ml by water. The blank sample with the same amount of acid was prepared with the same procedure.

**Secondary analysis of structure via circular dichroism spectra**. We used JASCO J-815 to obtain the circular dichroism spectra of NicX and its related proteins, at 20 °C, while the samples were 0.2 mg ml$^{-1}$ in 20 mM NaH₂PO₄-Na₂HPO₄ buffer (pH 7.4). We measured from 200 to 280 nm, with a cell length of 1.0 mm. We used the software CDPro to analyze the secondary structure compositions of the proteins.

**Dynamic pH titration**. The dynamic pH titrations were used to assess the dissociation constants (pKa) of DHP, in aqueous solutions. An ionic strength (I) of 0.1 mol L$^{-1}$ KCl was used to perform the titrations and a pH range of 4.0–12.0 was used to perform the measurements.

**MD simulation**. All MD simulations were performed by using the Amber software suite[46]. The initial protein structure used in the molecular dynamic simulation was constructed with the crystal structures (PDB: 7CUP、7CN3). The enzyme-substrate complexation was referred with the catalytic mechanisms of the Catechol dioxygenases and Rieske oxygenases[13,14]. The dioxygen tunnels toward the Fe center were analyzed with CAVER 3.0 program[47]. Both two water-coordination sites were tested for dioxygen displacement. In particular, the peroxide intermediate was docked into the active site based on the crystal structures with AutoDock 4 program[48]. Appropriate substrate conformations were selected for multiple MD simulations. Protonation states of titratable residues were assigned at pH 6.5 using the H++ web server[49], and Fe/substrate-binding residues were visually inspected then. The HF/6-31 G*//B3LYP/6-311 G* method was used to generate the substrate molecule in Gaussian 09, while the antechamber program was used to fit the RESP charge (restrained electrostatic potential charge)[50]. The python-based Metal Center Parameter Builder was used to build the force field of the iron center active site[51]. To prepare the topology and assess files of the enzyme-substrate complex, we used a cubic TIP3P water box (10 Å thick) from the surface of the complex, while sodium counter ion was used to neutralize the whole system. The particle mesh Ewald (PME) method was used to calculate the long-range electrostatic interactions in the MD simulation, while the SHAKE algorithm was used to constrain the hydrogen-involving bond lengths. We performed two minimization sequences, which relaxed the molecules of the solvents as well as the whole system. The temperature of the system increased from 0 to 300 K during 100 ps, at a collision frequency of 2 ps$^{-1}$ of Langevin dynamics. We calibrated the system for 50 ps and collected the trajectory with constant pressure and temperature (NPT). Finally, 25 ns production simulations without any restraint were performed under NPT conditions. An integration time step of 2 fs was utilized with structural snapshots being extracted every 1000 steps. The simulation trajectory was analyzed by the cpptraj in Amber tools18.

**QM calculations**. Geometrical snapshots from the enzyme-peroxide MD cluster were extracted as the pre-reaction states (PRS)[52] for the O-O and C-C bond breaking and forming, and were further subject to geometry optimization in Gaussian 09 program[53]. The quantum mechanical cluster model consisted of side chains of active site residues (His$^{105}$, Glu$^{177}$, His$^{189}$, His$^{265}$, Ser$^{302}$, His$^{318}$,and Asp$^{320}$) and the iron cation and peroxide intermediate, which added up to 99 atoms and bore one positive charge. The optimization process was carried out at the level of ωB97X-D functional and LANL2DZ (Fe) and 6-31 G(d) basis sets in aqueous solution with the steered molecular dynamics approach[54]. The triplet, quintet, and septet potential energy surfaces were scanned along the C-C and O-O reaction coordinates[55]. All stationary point structures were further optimized with the same level of theory. Vibrational frequency analyzes were performed to ensure local minima or first-order saddle points, and the free energies were calculated at 298 K (standard condition). In addition, the intrinsic reaction coordinates calculations were carried out to identify transition states and immediate reactants and products.

**Reporting summary**. Further information on research design is available in the Nature Research Reporting Summary linked to this article.

## Data availability

The data supporting the findings of this work are available within the paper and its Supplementary Information files or from the corresponding authors on reasonable request. Protein Data Bank (PDB): The coordinates and the structure factor amplitudes for the Se-NicX, NicX, and NicX complexed with ligands were deposited in Protein Data Bank under accession codes 7CNT, 7CUP, 7CN3; and have been released. Source data are provided with this paper.

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

## Acknowledgements

This work was supported by the grant of Ministry of Science and Technology of the People's Republic of China (2018YFA0901200), by the grant from Science and Technology Commission of Shanghai Municipality (17JC1403300), by the "Shuguang Program" (17SG09) supported by Shanghai Educational Development Foundation, by Shanghai Excellent Academic Leaders Program (20XD1421900) from Science and Technology Commission of Shanghai Municipality, and by the grants from National Natural Science Foundation of China (31422004, 31970041, and 21377085). We thank Dr. John H. Snyder, Prof. Jiahai Zhou, and Prof. Shuangjun Lin for assistance and discussion regarding this manuscript.

## Author contributions

H.T. and G.L. designed the experiments. G.L., F.H., and X.O. performed the experiments. G.L., H.T., and Y-L.Z. wrote the manuscript. G.L., Y-L.Z., H.T., and P.X. revised the manuscript. P.Z., G.L., and H.T. analyzed the data. H.T., Y-L.Z., and P.X. conceived the project.

## Competing interests

The authors declare no competing interests.
