## [Peer Review File · Nature Communications]

Reviewers' Comments:

Reviewer #1 (Remarks to the Author):

The authors report a crystal structure for dioxygenase NicX, which catalyses oxidative cleavage of a dihydroxypyridine substrate using a mononuclear iron (II) cofactor. There are some unusual features in the active site, notably the presence of four ligands for the iron (II) cofactor, but in essence the manuscript is the structure and site-directed mutagenesis data for active site residues, which seems not sufficiently transformative for Nature Communications, so I would recommend that the manuscript is more suitable for Nature Structural Biology.

Comments:

1. Regarding the proposed mechanism, the authors seem not to be fully aware of previous work on the catalytic mechanisms of the extradiol catechol dioxygenases, which are the most relevant class of enzyme. In particular, these enzymes are known to proceed via a semiquinone intermediate (formed by one electron transfer), which is likely to happen in this case, so there should be a semiquinone intermediate, and the two electron chemistry from structures 2 to 3 is incorrect. The 3rd structure is also incorrect, it should contain a carbonyl group at C-1.

2. Also there is literature work on the role of acid-base residues in the extradiol dioxygenases, indicating a role for an active site histidine in forming a substrate monoanion. It seems very likely that the nearby His105 does something similar in this enzyme.

3. From literature work on other iron-dependent dioxygenases, it is known that in general O₂ binding is to a vacant co-ordination site, it doesn't displace water. Therefore it is likely that a five co-ordinate species is formed, prior to O₂ binding, and perhaps this is the role of the conformational change observed.

4. line 217-8 There is no electron transport chain for this class of enzyme (unlike Rieske dioxygenases)

5. line 169 ferrous, not ferric

6. alpha-ketoglutarate (or 2-oxoglutarate)

Reviewer #2 (Remarks to the Author):

The authors present the first crystal structure of a non-heme dioxygenase that catalyzes ring cleavage of pyridine-based substrate. The study is notable in that the substrate does not coordinate the Fe(II) in contrast to other ring cleaving dioxygenases. Also the iron has an extra Ser ligand which is shown to be important via mutagenesis. The authors back up many of the claimed functions for metal ligands and active site residues by mutagenesis. There are several errors in the manuscript preparation (see below), particularly the fact that the reference numbers in the text do not all match up with the reference citations at the end. This made it difficult to follow the logic at times in the manuscript. The authors use kinetics to judge the result of a mutation in most cases. This is appropriate, but it would be good to provide some evidence that the protein folds correctly. For example, the loss of the poor iron ligand Ser would not seem sufficient in and of itself to result in the enzyme not binding Fe. One of the main points in the paper is the proposed roles of His105 and Leu104 in gating substrate entry and product release. This role is possible, but since the active site is always open via one or both routes, the alternative role also proposed for His105 in orienting the substrate seems more likely. In my opinion the gating role should be a point of discussion, not a result. The authors propose a mechanism for the enzyme based on the structure. However, the mechanisms shown is unlikely for several reasons.

One important reason is the lack of protons necessary to increase the reactivity of the Fe(III)-superoxo intermediate and/or break the O-O bond. In the extradiol dioxygenases there is a His in the active site that serves to remove a proton from the substrate and deliver it at the right time to activate the bound O₂. Is it possible at the N in the pyridine ring could serve a similar purpose? The authors propose the O₂ binding opposite the Ser ligand. This would not normally be the position in the Fe(II) enzymes where O₂ would bind opposite the Asp ligand. Electron transfer through the iron from this trans ligand would promote formation of the Fe(III)-superoxo bond. The authors draw the interesting connection to the Rieske dioxygenases in the discussion. These enzymes also bind the substrate away from the iron, and recent mechanistic studies suggest that the Fe(III)-superoxo intermediate can attack the aromatic substrate, more or less as shown in Fig. 6. However, this reaction is not favored and only occurs because the Rieske iron sulfur cluster in the Rieske dioxygenases can supply another electron immediately following the formation of the initial peroxo substrate radical intermediate. Also, the ring is not cleaved by these enzymes and the iron ends up in the Fe(III) state. If there is something special about the pyridine-based substrate that allows it to provide additional electrons for the ring cleaving reaction, it would be good to point it out in the discussion. Otherwise, this part seems too speculative. That said, I think it has potential with more elaboration. There are some things that remain difficult to rationalize about this enzyme. In particular, how the O₂ binding is triggered. It is possible that it is triggered by solvent release from the iron caused by nearby binding of the substrate. If this is the case it can be observed in the structure, it should be pointed out. In the extradiol enzymes, direct ligation of the substrate triggers O₂ binding, presumably by charge donation decreasing the potential of the iron. This apparently does not occur in NicX, so any insight from the crystal structures would be valuable to include.

Some specific problems

Lines 32-34 – Poor sentence

Line 51 – “apo form” generally refers to the metal free enzyme. This problem is repeated throughout the manuscript.

Line 61 – vestibule

Line 88 – Rieske

Line 89 alpha-ketoglutarate dependent

Line 90 – There are several more categories – intradiol dioxygenases and gentisate dioxygenases are among them.

Line 91 – There is no diol in the substrate so it can't be called an extradiol dioxygenase

Line 100 – Several are known with 3 His residues like gentisate dioxygenase

Line 104 – Reference required for this sentence

Line 125 – The reference numbers do not correlate with the list at the end from this point on.

Line 146 – Confusing sentence

Line 155 – Leu104 is rotated

Line 162 – The site is always open via one route or both, so why are these residues termed gate keepers?

Line 166 – ferrous ion

Line 169 – ferrous

Line 199 – ferrous

Line 203 – “in the” is repeated

Line 225 – missing words after “to form”

Line 240 – Fig 6 appears before Fig 5

Line 246 – O₂ would normally bind opposite the Asp so that it can accept charge via a trans effect.

Line 247 – DHP is not shown bound to the iron so Fe can't mediate electron transfer to O₂.

Line 267 – Not true. Gentisate dioxygenase, cys dioxygenase and others use 3 His.

Line 268 – ferrous

Line 279 – There is no relationship to a porphyrin, in ligand geometry or the type. This sentence serves no purpose.

Line 286 – Indirect evidence at best

Line 293 – “have” instead of “was” also “thereby allowing”

Line 292 to 296 – run-on, poorly constructed sentence

Line 308 – Rieske

Line 320 - hydroquinone

Reviewer #3 (Remarks to the Author):

In this manuscript the first structure from the NicX family of non-heme Fe(II) dioxygenase is determined and has an iron ion coordinated by 4 residues (including unusually a Ser) and 2 waters. In addition, the authors soaked substrate into the crystals in the presence of reductant, and found 4 of the subunits of the homohexameric enzyme contained bound substrate and the remaining 2 subunits contained product. To examine the roles of residues in the active site that interact with the metal, substrate and product, they use site-directed mutagenesis coupled to metal content and enzyme activity assays. In the resting enzyme and the product complex there are two distinct channels to the active site, but in the Michaelis complex two residues (L104 and H105) reorientate to effectively close one of the channels. Overall the experimental data presented have been appropriately modeled and handled, but the authors present only one interpretation of the structural changes that imply this interpretation is definitive. This is not the case and other interpretations are possible (I suggest some things to consider below). In addition, a clear figure showing all the key active site amino acids and the relative positional changes at the active site is not included. As such, I cannot create a full visual image of the juxtapositions in the active site to enable multiple structural mechanistic possibilities to be examined: Essentially the discussion is not chemically rigorous and a bit naive. A single mechanism is proposed with no data on catalytic intermediates (structural, pre-steady state, spectroscopic or otherwise). What is the mechanism presented based on (PnpCD, other ring-cleaving non-heme Fe(II) dioxygenases)? Looking at the mechanisms of different ring-cleaving dioxygenases, are there alternative mechanisms that might be proposed? As this is a new fold and active site environment for non-heme iron (II) dioxygenases, there is no compare-and-contrast analysis between the proposed NicX mechanism and the mechanisms of the other classes in the manuscript. The oxidation states of the irons in the structures are not addressed, and there is no accompanying spectroscopic evidence (e.g EPR, XAS) for intermediates etc. that would support their speculative mechanism. As such, this is a very unsatisfying manuscript, particularly for bioinorganic chemists and non-heme iron enzymologists.

(1) "Vestibule" is not a good term as it implies a side chamber or channel off another channel. I think "channel" or "tunnel" are more accurate.

(2) L61 and throughout the manuscript. The authors claim definitively which channels the substrate and product enter and exit: they present no data that supports this.

(3) L104. A reference is required for the previous work that demonstrated Ser302 was a ligand. Was this determined through mutagenesis and spectroscopic studies?

(4) L107 and throughout the manuscript. The term "apo" is incorrectly used to describe the resting holo-NicX. Apo would refer to the protein without iron bound. Use "resting" instead.

(5) L122. Why was molecular replacement necessary? The crystals appear essentially isomorphous.

(6) L123. The use of "monomer" is incorrect throughout; the authors mean "subunit" as NicX is a hexameric enzyme.

(7) L149. Fig. 2B should be Fig. 2C.

(8) The authors should describe the metal geometry and coordination distances. Ser is a very unusual transition metal ligand. Are there other non-heme iron enzymes that have a Ser ligand? The authors do not discuss iron oxidation state, and the crystal preparative conditions do not seem to have been rigorously controlled. In the Se-Met resting NicX structure the average iron B-factor suggests that iron occupancy is not 100% (the complex structure looks higher occupancy). From the conditions used to prepare the crystals the iron was probably ferric but may have been reduced in the x-ray beam. This is not discussed. For the complex dithionite was present when substrate was added, and clearly some turnover has occurred in the crystals. However, in the presence of oxygen and soak times of up to 60 minutes the reductant may have oxidized along with the iron. Ferric and ferrous iron geometries and coordination can be different, and the authors

are not rigorously controlling and checking oxidation state. Minimally, they need to state that they do not actually know what oxidation state they are observing in their crystal structures.

(9) The closing off of one channel to the active site in the DHP bound subunits is unlikely to have anything to do with substrate entry or product exit. The channel is remodeled by substrate after it has entered into the active site, and both channels become available again once product is formed. The authors present no data that define the paths of substrate entry and product exit. The remodeling is likely to be required for either productive oxygen binding and/or protecting/structuring the Michaelis complex for chemistry. Oxygen is hydrophobic: Does the closing of the channel create a hydrophobic path that would direct the oxygen to the iron? Is the mechanism ordered such that DHP must bind first for oxygen to be activated? Is the movement of L104/H105 simply stabilizing the optimal position of DHP to initiate the chemistry?

(10) L172. Table S1 should be referred to at the end of the sentence.

(11) L174. Table S2 should be referred to at the end of the sentence.

(12) L180-181. The distances reported are different from the values in Fig. 4A.

(13) L190. In writing about the kinetic parameters of the variants, the authors incorrectly use "activity" throughout when they mean K_{cat} or turnover.

(14) L196. Fig. 4B should be referred to at the end of the sentence.

(15) L199. Table 3 should be Table S2.

(16) L199-201. There is a mixing of oxidation states in this sentence in which the first half of the sentence refers to binding ferric iron (which is likely the correct oxidation state in the ICP-MS) so it does not directly prove the ability of the R293A variant to bind ferrous iron. In addition, Asp is a perfect ligand for iron without Arg, and so the role of Arg may be primarily steric to position the Asp correctly for ligation (although this does not preclude the possibility of an electrostatic role promoting the chemical mechanism).

(17) L204. Fig. 3 should be referred to at the end of the sentence.

(18) L205. Add "in the DHP bound subunits" to the end of the sentence.

(19) L207. The distance is labeled as 3.0 angstroms in Fig. 4C.

(20) L281. Reference 19 is not the correct reference as there is nothing in that paper regarding the role of Glu248. I think they mean Liu, S., Su, T., Zhang, C., Zhang, W.M., Zhu, D., Su, J., Wei, T., Wang, K., Huang, Y., Guo, L., Xu, S., Zhou, N.Y., Gu, L. (2015) *J Biol Chem* 290 24547-24560.

(21) L227. "(2.4 and 2.6 Å)" only references the distances from the two carboxylate O of E177. I believe it should be "(His, 2.8 angstroms; Glu, 2.4 and 2.6 angstroms)".

(22) L240. Fig. 6 is referenced before Fig. 5 in the text. Reverse the numbering.

(23) Include a figure with 2 panels that clearly shows all the active site side-chains whose roles were probed by mutagenesis, along with the metal for both a DHP and NFM bound subunit from the same orientation long with a third panel of the overlay. These can be in stereo if it is too crowded. I think such a figure really needs to be in the main manuscript.

(24) L268. "ferric" should be "ferrous".

(25) L293. "was disappeared, thereby" I think should be "are lost causing".

(26) L326. Reference Table S1 at the end of this sentence.

(27) L384. What was the ferric compound used in the ICP-MS assay?

(28) L397. "cryofreezing" should be "cryocooling".

(29) Table 1. The B-factors for the two structural models are quite high, but are reflective of the overall Wilson B-factors for the two structures. Include the Wilson B-factors in Table 1.

(30) Fig. 2A. I can only see 5 chain colors in the figure. Make the 6th subunit color more distinct so it can easily be seen.

(31) Fig. 3. Panels C and D are not informative. Replace these with panels suggested in point (23).

(32) L590-91. What is the grey mesh in the video?

Prof. Carrie M. Wilmot,
University of Minnesota, Twin Cities, USA

Reviewers' comments:

Response to Reviewer 1

REVIEWER #1

The authors report a crystal structure for dioxygenase NicX, which catalyses oxidative cleavage of a dihydroxypyridine substrate using a mononuclear iron (II) cofactor.

There are some unusual features in the active site, notably the presence of four ligands for the iron (II) cofactor, but in essence the manuscript is the structure and site-directed mutagenesis data for active site residues, which seems not sufficiently transformative for Nature Communications, so I would recommend that the manuscript is more suitable for Nature Structural Biology.

Response: Thanks for the encouragement and guidance about how to improve our study and manuscript.

Comments:

(1) Regarding the proposed mechanism, the authors seem not to be fully aware of previous work on the catalytic mechanisms of the extradiol catechol dioxygenases, which are the most relevant class of enzyme. In particular, these enzymes are known to proceed via a semiquinone intermediate (formed by one electron transfer), which is likely to happen in this case, so there should be a semiquinone intermediate, and the two electron chemistry from structures 2 to 3 is incorrect. The 3rd structure is also incorrect, it should contain a carbonyl group at C-1.

Response: Yes, thanks the help here; we have revised the related sentences. Further, we have now rigorously considered alternative pathways similar to the extradiol

catechol dioxygenases, wherein the dioxygen occupies the equatorial position and the DHP molecule is in the apical position (Pathway I in Figure 5). The two reactant ligands in Pathway I apparently bind at the active site more strongly than those in Pathway II, and appear similar to the mechanisms known for extradiol catechol dioxygenases; however, the specific substrate orientation in the co-crystal structure motivated us to reconsider the possibility of Pathway II. First, consider that the DHP molecule has to shift down by $\sim 4 \text{ \AA}$ to bind the metal and contact the equatorial dioxygen in Pathway I (Fig. S3); importantly, this would require disturbing the specific hydrogen bond network observed in the crystal structure. Second, the first 1-electron transfer of the DHP substrate seems to occur at the moiety of O-C=N based on the pK_a ($pK_{a1} = 8.56$) and Fukui f values (Fig. 5), which would require the dioxygen to absorb 1e from the ferrous center. Third, the O-O cleavage generally needs a proton donor, the highest-ranking of which is His¹⁰⁵ in Pathway I or Asp³²⁰ (or N-H) in Pathway II. Aspartic acid should be more likely than Histidine to donate a proton. Nevertheless, N-H---O hydrogen bonding reorganization is also possible, and both two pathways (with equatorial or apical dioxygen attack) need further investigation. We consider these ideas in detail in our comprehensively reworked result/discussion in the revised manuscript.

Figure 5. The two possible pathways in NicX-catalyzed DHP oxidation. Pathway I, dioxygen takes the opposite position of Asp³²⁰, DHP is coordinated with the ferrous ion; Pathway II, dioxygen takes the opposite position of Ser³⁰², DHP is not directly coordinated with the ferrous ion.

Figure S3. The calculated structures of critical peroxide intermediates in pathways IA (A), IB (B), and II (C), constructed with the crystal structures with substrate DHP and product NFM.

(2). Also there is literature work on the role of acid-base residues in the extradiol

dioxygenases, indicating a role for an active site histidine in forming a substrate monoanion. It seems very likely that the nearby His105 does something similar in this enzyme.

Response: Thank you for this thought-provoking advice. Based on our structure and experimental results. We speculate about two possible functions for the conformational change for Leu¹⁰⁴-His¹⁰⁵. The first is the aforementioned creation of a hydrophobic path that could greatly facilitate the direct delivery of a dioxygen molecule to the iron center (Figure S2). The second potential function of the conformational change could be to guide and bind the DHP substrate molecule to a pre-reaction position; this may facilitate activation of the dioxygen and promote reaction initiation. Our site-directed mutagenesis results showing that NicX^{C76Q} and NicX^{C76E} variants are inactivated provides further (albeit indirect) evidence of the contribution of Leu¹⁰⁴ and His¹⁰⁵ in orienting the substrate DHP.

Figure S2. A conformational change of Leu¹⁰⁴-His¹⁰⁵ create a hydrophobic path that goes straight to the active center of ferrous ion. A, In resting subunits or NFM bound subunits, the hydrophobic path is blocked by residue Leu¹⁰⁴, the dotted box zoomed in shows pockets made of hydrophobic amino acids. B, In DHP bound subunits,

conformational change of Leu¹⁰⁴-His¹⁰⁵ result in hydrophobic pockets connect with the active center of ferrous ion.

Figure 4. A conformational change of L104 and H105. *A*, In NFM bound subunits, Cys⁷⁶ is close to Leu¹⁰⁴ at a distance of 4.9 Å, away from His¹⁰⁵ at a distance of 7.7 Å. *B*. In DHP bound subunits, Cys⁷⁶ is close to His¹⁰⁵ at a distance of 4.1 Å, away from Leu¹⁰⁴ at a distance of 7.7 Å.

(3). From literature work on other iron-dependent dioxygenases, it is known that in general O₂ binding is to a vacant co-ordination site, it doesn't displace water.

Therefore it is likely that a five co-ordinate species is formed, prior to O₂ binding, and perhaps this is the role of the conformational change observed.

Response: Thank you these helpful notes. We rewrote the mechanism section in the revised version. See page 12-13 line 233-263.

(4). line 217-8 There is no electron transport chain for this class of enzyme (unlike Rieske dioxygenases)

Response: Thank you for this very helpful guidance; we have deleted the sentence from the revised manuscript.

➤ **Solution:** We deleted the sentence in the revised version.

(5). line 169 ferrous, not ferric

Response: Thanks for highlighting this error.

➤ **Solution:** We corrected it throughout the revised manuscript.

(6). alpha-ketoglutarate (or 2-oxoglutarate)

Response: Thank you for kindly reminding. Corrected.

➤ **Solution:** We corrected this in the revised manuscript.

Let us again express our appreciation for the encouragement and excellent helpful guidance about how to improve our study. Many thanks!

Response to Reviewer 2

REVIEWER #2

The authors present the first crystal structure of a non-heme dioxygenase that catalyzes ring cleavage of pyridine-based substrate. The study is notable in that the substrate does not coordinate the Fe(II) in contrast to other ring cleaving dioxygenases. Also the iron has an extra Ser ligand which is shown to be important via mutagenesis. The authors back up many of the claimed functions for metal ligands and active site residues by mutagenesis. There are several errors in the manuscript preparation (see below), particularly the fact that the reference numbers in the text do not all match up with the reference citations at the end. This made it difficult to follow the logic at times in the manuscript.

Response: Thank you very much for providing us with important comments and guidance that have been foundational as we have improved our study and revised its manuscript. Regarding this comment specifically, we have carefully checked the serial numbers of the references in the revised manuscript.

➤ **Solution:** We have carefully checked the serial numbers of the references in the revised manuscript.

(1). The authors use kinetics to judge the result of a mutation in most cases. This is appropriate, but it would be good to provide some evidence that the protein folds correctly. For example, the loss of the poor iron ligand Ser would not seem sufficient in and of itself to result in the enzyme not binding Fe.

Response: Thanks for the reviewer's professional suggestion here. We have now performed circular dichroism experiments on these mutants that lost their ability to bind Fe. The circular dichroism analysis showed that the mutants do not affect protein folding (See Table S3, Figure S1).

➤ **Solution:** Guided by the reviewer's helpful suggestions, we performed circular dichroism (CD) spectra to determine whether mutations affect protein folding. Circular dichroism (CD) spectra showed that mutants fold in the native form. Revised Table S3 and Figure S1 are as follows:

Supplemental Table S3. Secondary structure analysis of NicX and its mutants based on circular dichroism spectra

Enzymes	α -helix	β -sheet	Random
WT	9.60%	57.00%	33.40%
H265A	10.20%	57.00%	32.80%
R293A	10.60%	57.20%	32.20%
S302A	10.40%	57.20%	32.40%
H318A	10.30%	57.50%	32.20%
D320A	9.80%	57.30%	32.90%

Supplemental Figure S1. Circular dichroism spectra of NicX and its mutants.

(2). One of the main points in the paper is the proposed roles of His105 and Leu104 in gating substrate entry and product release. This role is possible, but since the active site is always open via one or both routes, the alternative role also proposed for His105 in orienting the substrate seems more likely. In my opinion the gating role should be a point of discussion, not a result.

Response: Thanks for the reviewer's professional and helpful suggestions which led us gain several new insights into the conformational change of L104 and H105. Following the reviewer's comments, we agree that the role of L104 and H105 should be a point of discussion; this has been moved to the revised discussion section (Page14 line274-293).

In both the substrate/product binding states, we find that C76 is closer to Leu104 and His105, so C76 in channel II was selected for mutation (Figure 4), the point mutations

of C76Q or C76E completely abolished the enzymatic activity of NicX; in addition, C76A retained the capacity of transforming DHP into NFM. We speculate that the bulky residues (Glutamine or Glutamic acid) take up more space than Cysteine, and affect the range of movement of L104 and H105, potentially disrupting the ability to guide and stabilize the substrate to the appropriate position to initiate the reaction. Our data for the activity of the NicX variants support the review's view about the function of H105 is in orienting the substrate. This is shown in the results section (Page 11 line 215-231), and is discussed in detail in the discussion of the revised manuscript (Page 14 line 274-293).

➤ **Solution:** We moved description of potential roles of His105 and Leu104 to the discussion section. And following the reviewer's comments, C76 in channel II that was chosen for mutation into bulky residues. The C76Q or C76E mutations completely abolished the enzymatic activity of NicX; in addition, C76A retained the capacity to transform DHP into NFM. This result supports that bulky residues lead to enzyme inactivation by affecting the range of movement of Lue104 and His105, findings supporting the review's view.

Figure 4. A conformational change of L104 and H105. *A*, In NFM bound subunits, Cys⁷⁶ is close to Leu¹⁰⁴ at a distance of 4.9 Å, away from His¹⁰⁵ at a distance of 7.7 Å. *B*. In DHP bound subunits, Cys⁷⁶ is close to His¹⁰⁵ at a distance of 4.1 Å, away from Leu¹⁰⁴ at a distance of 7.7 Å.

(3) The authors propose a mechanism for the enzyme based on the structure. However, the mechanisms shown is unlikely for several reasons. One important reason is the lack of protons necessary to increase the reactivity of the Fe(III)-superoxo intermediate and/or break the O-O bond. In the extradiol dioxygenases there is a His in the active site that serves to remove a proton from the substrate and deliver it at the

right time to activate the bound O₂. Is it possible at the N in the pyridine ring could serve a similar purpose? The authors propose the O₂ binding opposite the Ser ligand. This would not normally be the position in the Fe(II) enzymes where O₂ would bind opposite the Asp ligand. Electron transfer through the iron from this trans ligand would promote formation of the Fe(III)-superoxo bond. The authors draw the interesting connection to the Rieske dioxygenases in the discussion. These enzymes also bind the substrate away from the iron, and recent mechanistic studies suggest that the Fe(III)-superoxo intermediate can attack the aromatic substrate, more or less as shown in Fig. 6. However, this reaction is not favored and only occurs because the Rieske iron sulfur cluster in the Rieske dioxygenases can supply another electron immediate following the formation of the initial peroxo substrate radical intermediate. Also, the ring is not cleaved by these enzymes and the iron ends up in the Fe(III) state. If there is something special about the pyridine-based substrate that allows it to provide additional electrons for the ring cleaving reaction, it would be good to point it out in the discussion. Otherwise, this part seems too speculative. That said, I think it has potential with more elaboration.

Response: Thanks for the reviewer's professional suggestions. We have considered alternative pathways similar to the extradiol catechol dioxygenases, by which dioxygen occupies at the equatorial position and DHP molecule at the apical position (Pathway I in Figure 5). The two reactant ligands in Pathway I look like binding the active site more strongly than those in Pathway II and the similar mechanism as the extradiol catechol dioxygenase takes place; however, the specific substrate orientation

in the co-crystal structure let us to reconsider the possibility of Pathway II. First of all, DHP molecule has to shift down by $\sim 4 \text{ \AA}$ to bind the metal and contact the equatorial dioxygen in Pathway I, which would disturb the specific hydrogen bond network observed in the crystal structure (Figure S3). Second, the first 1-electron transfer of the DHP substrate seems to occur at the moiety of O-C=N based on the pK_a ($pK_{a1} = 8.56$) and Fukui function f (Fig. 5), which requires dioxygen molecule to absorb 1e from the ferrous center ahead. Third, the rate-limiting O-O cleavage generally needs a proton donor, the highest possible for which is either His¹⁰⁵ in Pathway I or Asp³²⁰ (or N-H) in Pathway II. Aspartic acid seems much better than Histidine to donate a proton, whereas the N-H---O hydrogen bonding reorganization is also possible. Indeed, both the two pathways with equatorial and apical dioxygen attack need further investigation.

In particular, the DHP molecule likely adopts monoanionic form ($pK_{a1}=8.56$) in the hydrogen bonding network. Frontier molecular orbital analysis and Fukui function f (0.148) suggests that the C6 position is the most vulnerable to electrophile and thereby for the nascent C-O bond with dioxygen via 1e transfer to initialize the reaction. Given that the pyridinyl N-H can donate a proton in the consequent O-O cleavage, the substrate specificity might also be resulted from electronic reasons. This is shown in the results section (Page 12-13 line 233-263), discussion section (Page 15-16 line 311-329).

Figure 5. The two possible pathways in NicX-catalyzed DHP oxidation. Pathway I, dioxygen takes the opposite position of Asp³²⁰, DHP is coordinated with the ferrous ion; Pathway II, dioxygen takes the opposite position of Ser³⁰², DHP is not directly coordinated with the ferrous ion.

Figure S3. The calculated structures of critical peroxide intermediates in pathways IA (A), IB (B), and II (C), constructed with the crystal structures with substrate DHP and product NFM.

(4) There are some things that remain difficult to rationalize about this enzyme. In

particular, how the O₂ binding is triggered. It is possible that it is triggered by solvent release from the iron caused by nearby binding of the substrate. If this is the case it can be observed in the structure, it should be pointed out. In the extradiol enzymes, direct ligation of the substrate triggers O₂ binding, presumably by charge donation decreasing the potential of the iron. This apparently does not occur in NicX, so any insight from the crystal structures would be valuable to include.

Response: Careful examination of the crystal structure as motivated by this comment led to our surprising finding of a hydrophobic path when channel II was closed in the E subunit. This hydrophobic path is blocked by Leu104 in the resting state or product NFM binding state; we interpret this to mean that the new hydrophobic path could be responsible for facilitating oxygen entry, as induced by the conformational change of L104-H105 initiated upon substrate binding.

Figure S2. A conformational change of Leu¹⁰⁴-His¹⁰⁵ create a hydrophobic path that goes straight to the active center of ferrous ion. A, In resting subunits or NFM bound subunits, the hydrophobic path is blocked by residue Leu¹⁰⁴, the dotted box zoomed in shows pockets made of hydrophobic amino acids. B, In DHP bound subunits, conformational change of Leu¹⁰⁴-His¹⁰⁵ result in hydrophobic pockets connect with the active center of ferrous ion.

Some specific problems

(1) Lines 32-34 – Poor sentence

Response: Thank you for the helpful suggestion. We have reworked this sentence for clarity.

➤ **Solution:** We reworded the sentence, shown on Page 3 line 54-56.

(2) Line 51 – “apo form” generally refers to the metal free enzyme. This problem is repeated throughout the manuscript.

Response: Thank you for this helpful guidance; we have corrected this error throughout the manuscript.

➤ **Solution:** “apo form” was replaced by “resting form” in the revised manuscript.

(3) Line 61 – vestibule

Response: Thank you for the helpful suggestion. We have revised the sentence.

➤ **Solution:** “vestibule” was replaced by “channel” in the revised manuscript.

(4) Line 88 – Rieske

Response: Thank you for the helpful suggestion. We have revised the sentence.

➤ **Solution:** “rieske” was replaced by “Rieske” in revised version.

(5) Line 89 alpha-ketoglutarate dependent

Response: Thank you for the helpful suggestion; we have corrected this in the revised manuscript.

➤ **Solution:** “a-ketoglutaratedependent” was replaced by “alpha-ketoglutarate dependent” in the revised manuscript.

(6) Line 90 – There are several more categories – intradiol dioxygenases and gentisate dioxygenases are among them.

Response: Thank you for this important suggestion. This content has been revised.

➤ **Solution:** We consulted the previous literature and rewrote the related sentences. Please see as follows.

P4 line 75: “These enzymes can be classified into several different groups based on their structural characteristics, reactivity, and specific requirements for catalysis, including: (I) Catechol (intradiol and extradiol) dioxygenases, (II) Rieske oxygenases, (III) Alpha-ketoglutarate dependent enzymes, (IV) Cysteine dioxygenases, (V) Pterin-dependent hydroxylases, among 2-Hydroxyethylphosphonate dioxygenases.”

(7) Line 91 – There is no diol in the substrate so it can't be called an extradiol dioxygenase

Response: Thank you for the helpful suggestion. We have revised the sentence.

➤ **Solution:** “A phylogenetic analysis of non-heme Fe(II) dioxygenases indicted that NicX is a member of a new subclass of the non-heme iron dependent dioxygenases ” in Page 5 line 81-82.

(8) Line 100 – Several are known with 3 His residues like gentisate dioxygenase

Response: Thank you for the helpful suggestion. We examined the previous literature and rewrote the sentences.

➤ **Solution:** We added the sequence “In addition, a His/His/His triad coordinates Fe(II) has been found in cysteine dioxygenases and gentisate dioxygenase (22–24).” in Page 5 line 93-94.

(9) Line 104 – Reference required for this sentence

Response: Thank you for pointing out the need for a reference here. This sequence was reworded in Page 5 line 95-96 “Interestingly, NicX Ser³⁰² was found to coordinate the iron(II) ion; such a metal ion-interacting serine residue has been reported in a dialkylglycine decarboxylase (25)”. Reference was showed below.

[Reference]

Toney, M., Hohenester, E., Cowan, S., & Jansonius J. Dialkylglycine decarboxylase structure: bifunctional active site and alkali metal sites. *Science*. **261**, 756–759 (1993)

➤ **Solution:** A related reference was added.

(10) Line 125 – The reference numbers do not correlate with the list at the end from this point on.

Response: Thank you for the bringing these errors to our attention. We have carefully corrected this issue in the revised manuscript.

➤ **Solution:** We have carefully checked the numbers of the references in the revised manuscript.

(11) Line 146 – Confusing sentence

Response: Thank you for the helpful suggestion, and we have removed this sentence.

➤ **Solution:** The confusing sentence is not present in the revised manuscript.

(12) Line 155 – Leu104 is rotated

Response: Thank you for the helpful suggestion. We have revised the sentence.

➤ **Solution:** “rotates” has been replaced by “rotated” in the revised manuscript.

(13) Line 162 – The site is always open via one route or both, so why are these residues termed gate keepers?

Response: Thank you for this guidance. We have revised the related sentence.

➤ **Solution:** Kindly note that all mention of the L104-H105 residues “gate keepers” have been removed from the revised manuscript.

(14) Line 166 – ferrous ion

Response: Thank you for the great suggestion. Corrected in revised version.

➤ **Solution:** “ferrous iron” was replaced by “ferrous ion”.

(15) Line 169 – ferrous

Response: Thank you for kindly reminding. Corrected in revised version.

➤ **Solution:** “ferric” was replaced by “ferrous”.

(16) Line 199 – ferrous

Response: Thank you for the great suggestion. Corrected in revised version.

➤ **Solution:** “ferric” was replaced by “ferrous”.

(17) Line 203 – “in the” is repeated

Response: Thank you for the great suggestion. We have revised the sentence.

➤ **Solution:** We deleted one of instances of ‘in the’.

(18) Line 225 – missing words after “to form”

Response: Thank you for the great suggestion. We have revised the sentence.

➤ **Solution:** The sequence in Page 10 line189-191. “Analysis of the ring-open product NFM in the active sites of the A and B subunits of the complex structure revealed that the pyridine ring of DHP has been cleaved between the C5^{DHP} and C6^{DHP} carbons to form the product NFM.”.

(19) Line 240 – Fig 6 appears before Fig 5

Response: Thank you for the great suggestion. Corrected in revised version.

➤ **Solution:** The pictures have been redrawn and we rearranged the order of the pictures in the revised version.

(20) Line 246 – O2 would normally bind opposite the Asp so that it can accept charge via a trans effect.

Response: Thank you for the great suggestion. We have now rigorously considered alternative pathways similar to the extradiol catechol dioxygenases, wherein the dioxygen occupies the equatorial position (opposite the Asp³²⁰) and the DHP molecule is in the apical position (Pathway I in Figure 5). The two reactant ligands in Pathway I apparently bind at the active site more strongly than those in Pathway II, and appear similar to the mechanisms known for extradiol catechol dioxygenases; however, the specific substrate orientation in the co-crystal structure motivated us to reconsider the possibility of Pathway II. First, consider that the DHP molecule has to shift down by ~4 Å to bind the metal and contact the equatorial dioxygen in Pathway I (Fig. S3); importantly, this would require disturbing the specific hydrogen bond network observed in the crystal structure. Second, the first 1-electron transfer of the DHP substrate seems to occur at the moiety of O-C=N based on the pK_a ($pK_{a1} = 8.56$) and Fukui f values (Fig. 5), which would require the dioxygen to absorb 1e from the ferrous center. Third, the O-O cleavage generally needs a proton donor, the highest-ranking of which is His¹⁰⁵ in Pathway I or Asp³²⁰ (or N-H) in Pathway II. Aspartic acid should be more likely than Histidine to donate a proton. Nevertheless, N-H---O hydrogen bonding reorganization is also possible, and both two pathways

(with equatorial or apical dioxygen attack) need further investigation. This is shown in the results section (Page 12-13 line 233-263), discussion section (Page 15-16 line 311-329).

Figure 5. The two possible pathways in NicX-catalyzed DHP oxidation. Pathway I, dioxygen takes the opposite position of Asp³²⁰, DHP is coordinated with the ferrous ion; Pathway II, dioxygen takes the opposite position of Ser³⁰², DHP is not directly coordinated with the ferrous ion.

Figure S3. The calculated structures of critical peroxide intermediates in pathways IA (A), IB (B), and II (C), constructed with the crystal structures with substrate DHP and

product NMF.

(21) Line 247 – DHP is not shown bound to the iron so Fe can't mediate electron transfer to O₂.

Response: Thank you for the great suggestion. We don't know much about electron transfer to O₂. Further investigation on electron transfer to O₂ is needed.

(22) Line 267 – Not true. Gentsate dioxygenase, cys dioxygenase and others use 3 His.

Response: Thank you for the correcting this error. We have remove this wrong sentence in the revised version.

(23) Line 268 – ferrous

Response: Thank you for the great suggestion. Corrected in revised version.

➤ **Solution:** “ferric” replaced by “ferrous”.

(24) Line 279 – There is no relationship to a porphyrin, in ligand geometry or the type. This sentence serves no purpose.

Response: Thank you for this guidance; we have removed this sentence from the revised manuscript.

Solution: We removed this sentence from the revised manuscript.

(25) Line 286 – Indirect evidence at best

Response: Thank you for the great suggestion. We have revised the sentence.

➤ **Solution:** “direct” replaced by “indirect”.

(26) Line 293 – “have” instead of “was” also “thereby allowing”

Response: Thank you for the great suggestion. We have rewritten the discussion section, sentences from Line 292 to 296 this part which were removed in the revised version.

➤ **Solution:** The sentence was removed in revised version.

(27) Line 292 to 296 – run-on, poorly constructed sentence

Response: Thank you for the great suggestion. We have written the discussion section, sentences from Line 292 to 296 this part which were removed in the revised version.

➤ **Solution:** The sentence was removed in revised version.

(28) Line 308 – Rieske

Response: Thank you for this correction. Note that although the related content is no longer present in our comprehensively reworked discussion section, we now understand how to use this term properly and are grateful for the clarification.

(29) Line 320 – hydroquinone

Response: Thank you for the great suggestion. We have written the discussion section, the sentence was removed in the revised version, but we have learned how to use words correctly. Learned to use words correctly.

We would like to take this opportunity to thank the reviewer for the careful and extremely helpful guidance about how to improve our study and manuscript.

Response to Reviewer 3

REVIEWER #3

In this manuscript the first structure from the NicX family of non-heme Fe(II) dioxygenase is determined and has an iron ion coordinated by 4 residues (including unusually a Ser) and 2 waters. In addition, the authors soaked substrate into the crystals in the presence of reductant, and found 4 of the subunits of the homohexameric enzyme contained bound substrate and the remaining 2 subunits contained product. To examine the roles of residues in the active site that interact with the metal, substrate and product, they use site-directed mutagenesis coupled to metal content and enzyme activity assays. In the resting enzyme and the product complex there are two distinct channels to the active site, but in the Michaelis complex two residues (L104 and H105) reorientate to effectively close one of the channels. Overall the experimental data presented have been appropriately modeled and handled, but the authors present only one interpretation of the structural changes that imply this interpretation is definitive. This is not the case and other interpretations are possible (I suggest some things to consider below).

Response: Thanks for the reviewer's encouragements and professional suggestions. With this exceedingly helpful guidance, we have now totally reworked several aspects of our study; further we now present a much more carefully considered (and properly cited) introduction and discussion.

(1) In addition, a clear figure showing all the key active site amino acids and the

relative positional changes at the active site is not included. As such, I cannot create a full visual image of the juxtapositions in the active site to enable multiple structural mechanistic possibilities to be examined.

Response: Thanks for the reviewer's professional and crucial suggestions. We revised the figure as suggested.

➤ **Solution:** We revised the figure as follows:

Figure 3. Active site of NicX. *A*, Active site of NicX with bound DHP. The carbon atoms of DHP are in yellow. Green, red, blue and purple represent carbon, oxygen, nitrogen, and Fe atoms respectively. *B*, Active site of NicX with bound NFM. The carbon atoms of NFM are in yellow. Cyan, red, blue and purple represent carbon, oxygen, nitrogen and Fe atoms respectively. *C*, A stereoview of the DHP bound subunit (green) and NFM bound subunit (cyan), the superimposition was done on the whole subunit. Iron and solvent molecules are shown as purple and red spheres, respectively. Distances are given in angstroms.

(2) Essentially the discussion is not chemically rigorous and a bit naive. A single mechanism is proposed with no data on catalytic intermediates (structural, pre-steady state, spectroscopic or otherwise). What is the mechanism presented based on (PnpCD, other ring-cleaving non-heme Fe(II) dioxygenases)? Looking at the mechanisms of different ring-cleaving dioxygenases, are there alternative mechanisms that might be proposed?

Response: Thanks for the reviewer's frank and helpful guidance here. We have now rigorously considered alternative pathways similar to the extradiol catechol dioxygenases, wherein the dioxygen occupies the equatorial position and the DHP molecule is in the apical position (Pathway I in Figure 5). The two reactant ligands in Pathway I apparently bind at the active site more strongly than those in Pathway II, and appear similar to the mechanisms known for extradiol catechol dioxygenases; however, the specific substrate orientation in the co-crystal structure motivated us to

reconsider the possibility of Pathway II. First, consider that the DHP molecule has to shift down by $\sim 4 \text{ \AA}$ to bind the metal and contact the equatorial dioxygen in Pathway I (Fig. S3); importantly, this would require disturbing the specific hydrogen bond network observed in the crystal structure. Second, the first 1-electron transfer of the DHP substrate seems to occur at the moiety of O-C=N based on the pK_a ($pK_{a1} = 8.56$) and Fukui f values (Fig. 5), which would require the dioxygen to absorb 1e from the ferrous center. Third, the O-O cleavage generally needs a proton donor, the highest-ranking of which is His¹⁰⁵ in Pathway I or Asp³²⁰ (or N-H) in Pathway II. Aspartic acid should be more likely than Histidine to donate a proton. Nevertheless, N-H---O hydrogen bonding reorganization is also possible, and both two pathways (with equatorial or apical dioxygen attack) need further investigation. We consider these ideas in detail in our comprehensively reworked result/discussion in the revised manuscript. This is shown in the results section (Page 12-13 line 233-263), discussion section (Page 15-16 line 311-329).

Figure 5. The two possible pathways in NicX-catalyzed DHP oxidation. Pathway I, dioxygen takes the opposite position of Asp³²⁰, DHP is coordinated with the ferrous ion; Pathway II, dioxygen takes the opposite position of Ser³⁰², DHP is not directly coordinated with the ferrous ion.

Figure S3. The calculated structures of critical peroxide intermediates in pathways IA (A), IB (B), and II(C), constructed with the crystal structures with substrate DHP and product NFM.

(3) As this is a new fold and active site environment for non-heme iron (II) dioxygenases, there is no compare-and-contrast analysis between the proposed NicX

mechanism and the mechanisms of the other classes in the manuscript.

Response: Thanks for the reviewer's helpful suggestions. In discussion section, we compared NicX with the classic extradiol catechol dioxygenases, shown in Page 16-17 line 330-350.

(4) The oxidation states of the irons in the structures are not addressed, and there is no accompanying spectroscopic evidence (e.g EPR, XAS) for intermediates etc. that would support their speculative mechanism. As such, this is a very unsatisfying manuscript, particularly for bioinorganic chemists and non-heme iron enzymologists.

Response: Thanks for the reviewer's guidance. We performed EPR and stopped-flow experiments, but unfortunately, we did not detect the oxidation states of the irons during the catalysis process, and obtained no spectroscopic evidence for intermediates. We were able to obtain a 130 K spectrum, but are presently unable to conduct low temperature assays. Please kindly see the following figures.

Figure A, 130K spectrum 1 min after mixing NicX-DHP complex with O₂ saturated buffer at 4°C. **B**, Single turnover of the preformed NicX-DHP complex upon exposure to O₂.

(5) “Vestibule” is not a good term as it implies a side chamber or channel off another channel. I think “channel” or “tunnel” are more accurate.

Response: Thank you for this fundamentally impactful suggestion about how to improve our manuscript! Following the reviewer's comment, "vestibule" has been replaced by "channel" throughout the revised manuscript, and we trust the reviewer will agree that our story and mechanistic interpretations are now much clearer.

➤ **Solution:** "vestibule" has been replaced by "channel" throughout the revised manuscript.

(6) L61 and throughout the manuscript. The authors claim definitively which channels the substrate and product enter and exit: they present no data that supports this.

Response: Thank you for the great suggestion. In the revised manuscript, the channel I and channel II content no longer include ideas about substrate entry and product release channels.

➤ **Solution:** We no longer emphasize substrate entry and product release channels in the revised version.

(7) L104. A reference is required for the previous work that demonstrated Ser302 was a ligand. Was this determined through mutagenesis and spectroscopic studies?

Response: Thank you for the great suggestion. This sequence was reworded in Page 5 line 95-96 "Interestingly, NicX Ser³⁰² was found to coordinate the iron(II) ion; such a metal ion-interacting serine residue has been reported in a dialkylglycine decarboxylase (25)"; the reference is shown below. Site mutagenesis, circular dichroism and ICP-MS assays were performed to determine Ser302 was a ligand in

this study.

➤ **Solution:** Related reference was added.

[Reference]

Toney, M., Hohenester, E., Cowan, S., & Jansonius, J. Dialkylglycine decarboxylase structure: bifunctional active site and alkali metal sites. *Science*. **261**, 756–759 (1993).

(8) L107 and throughout the manuscript. The term “apo” is incorrectly used to describe the resting holo-NicX. Apo would refer to the protein without iron bound. Use “resting” instead.

Response: Thanks for the reviewer’s professional suggestion. We have revised this diction throughout manuscript.

➤ **Solution:** “apo” was replaced by “resting” throughout the revised manuscript as suggested.

(9) L122. Why was molecular replacement necessary? The crystals appear essentially isomorphous.

Response: Thank you for this comment. Indeed, molecular replacement was not necessary; note that we used molecular replacement to help solve the complex structure.

➤ **Solution:** The sentence was reworded; see Page 6 line 115-116. “Subsequently, the crystal structures of NicX and the NicX-DHP-NFM complex were solved using coordinates of Se-Met NicX (Table 1).”

(10) L123. The use of “monomer” is incorrect throughout; the authors mean “subunit” as NicX is a hexameric enzyme.

Response: Thank you for the great suggestion. Corrected.

➤ **Solution:** “monomer” replaced by “subunit” throughout the manuscript.

(11) L149. Fig. 2B should be Fig. 2C.

Response: Thank you for the great suggestion. Sorry for making the mistake.

➤ **Solution:** We rearranged the order of the pictures.

(12) **A)** The authors should describe the metal geometry and coordination distances. **B)**

Ser is a very unusual transition metal ligand. Are there other non-heme iron enzymes

that have a Ser ligand? **C)** The authors do not discuss iron oxidation state, and the

crystal preparative conditions do not seem to have been rigorously controlled. In the

Se-Met resting NicX structure the average iron B-factor suggests that iron occupancy

is not 100% (the complex structure looks higher occupancy). **D)** From the conditions

used to prepare the crystals the iron was probably ferric but may have been reduced in

the x-ray beam. This is not discussed. For the complex dithionite was present when

substrate was added, and clearly some turnover has occurred in the crystals. However,

in the presence of oxygen and soak times of up to 60 minutes the reductant may have

oxidized along with the iron. Ferric and ferrous iron geometries and coordination can

be different, and the authors are not rigorously controlling and checking oxidation

state. Minimally, they need to state that they do not actually know what oxidation

state they are observing in their crystal structures.

Response: Thank you for the professional suggestion. We reply as follows according to your suggestions and questions

Response part A) We showed the metal coordination distances of DHP-/NFM-binding subunit in revised version. Shown as Fig 3A, 3B.

Figure 3. Active site of NicX. *A*, Active site of NicX with bound DHP. The carbon atoms of DHP are in yellow. Green, red, blue and purple represent carbon, oxygen, nitrogen and Fe atoms respectively. *B*. Active site of NicX with bound NFM. The carbon atoms of NFM are in yellow. Cyan, red, blue and purple represent carbon, oxygen, nitrogen and Fe atoms respectively.

Response part B) Based on our literature research, no any other non-heme iron enzymes have been reported to use serine as a metal ligand. However, Other metalloenzyme uses serine as a ligand, the corresponding literature is listed below.

[References]

M. Toney, E Hohenester, S. Cowan, & J. Jansonius. Dialkylglycine decarboxylase structure: bifunctional active site and alkali metal sites. *Science*. **261**, 756–759 (1993)

Response part C) When we prepared selenium-substituted crystals, We were focused on solving the phase problem rather than metal occupancy. We have now provided new diffraction data about the NicX structure in the revised version. Data are listed in Table 1.

TABLE 1 Data collection and refinement statistics

	SeMet-NicX	NicX	NicX in complex with DHP and NFM
Data collection			
Space group	P2₁2₁2	P2₁2₁2	P2₁2₁2
Wavelength (Å)	0.97918	0.97892	0.97918
Cell dimensions			
a, b, c (Å)	125.92, 144.13, 118.89	125.961, 143.75, 118.69	126.67, 145.51, 118.98
α , β , γ (°)	90.00, 90.00, 90.00	90.00, 90.00, 90.00	90.00, 90.00, 90.00
Number of molecules/asymmetric unit	6	6	6
Resolution range (Å) (outer shell)	50-2.28 (2.40-2.28)	50.00-2.00 (2.03-2.00)	50-2.20 (2.24-2.20)
Completeness (%) (outer shell)	99.2 (99.6)	100.0 (97.3)	99.8 (99.2)
Redundancy (outer shell)	6.0 (6.2)	12.3 (12.3)	12.8 (13.1)
Total observations	585113	1763979	1453302
Unique reflections	98134	142933	113279
Wilson B factor (Å ²)	50.13	22.91	27.51
R _{merge} (%) (outer shell)	9.1 (65.1)	10.1 (95.3)	10.6 (95.9)
I/ σ _I (outer shell)	8.7 (3.0)	25 (2.67)	24.4 (3.7)
Refinement			
Resolution range (Å)	30-2.28	47.41-2.00	27.59-2.20
R _{work} /R _{free} (%)	21.1/26.7	17.12/21.33	18.3/23.8
Average B-factors(Å ²)			
Protein, Metal ion, Water, Substrate/Product	67.3, 97.2, 55.1, -	27.75, 38.04, 40.28, -	29.52, 45.6, 38.8, 42.9/55.1
Root mean square deviations			
Bond angles (°), Bond lengths (Å)	0.014, 1.644	0.010, 1.625	0.008, 0.908
Number of atoms			
Protein/Substrate/Water	16371/0/241	16334/0/1514	16576/52/940
Ramachandran plot			
Most favored, Allowed, Disallowed (%)	95.1, 4.7, 0.1	95.81, 3.66, 0.53	95.38, 4.23, 0.38

Response part D) Actually, we do not know what oxidation state of iron which in crystal structures; this is now clearly stated in the revised version. See Page 8 line 161-163. “It should be noted that although crystals were soaked in the buffer contain Fe^{2+} , we were not able to experimentally determine the oxidation state of the iron in the crystal structures”.

(9) The closing off of one channel to the active site in the DHP bound subunits is unlikely to have anything to do with substrate entry or product exit. The channel is remodeled by substrate after it has entered into the active site, and both channels become available again once product is formed. The authors present no data that define the paths of substrate entry and product exit. The remodeling is likely to be required for either productive oxygen binding and/or protecting/structuring the Michaelis complex for chemistry. Oxygen is hydrophobic: Does the closing of the channel create a hydrophobic path that would direct the oxygen to the iron? Is the mechanism ordered such that DHP must bind first for oxygen to be activated? Is the movement of L104/H105 simply stabilizing the optimal position of DHP to initiate the chemistry?

Response: Thanks for the reviewer’s professional and crucial suggestions which let us gain new insights into the nature and consequences of the conformational change of Leu104 and His105.

1) Following the reviewer’s comment, we carefully examined the structure, and it was a surprise to find the creation of a hydrophobic path upon closing of channel II in the

E subunit (Figure S2). This hydrophobic path is blocked by L104 in the resting state or by the product in the NFM-binding state; this indicates that a new hydrophobic path is apparently induced by the conformational change of L104-H105. We discuss this in considerable detail in the revised manuscript.

Figure S2. A conformational change of Leu¹⁰⁴-His¹⁰⁵ create a hydrophobic path that goes straight to the active center of ferrous ion. A, In resting subunits or NFM bound subunits, the hydrophobic path is blocked by residue Leu¹⁰⁴, the dotted box zoomed in shows pockets made of hydrophobic amino acids. B, In DHP bound subunits, conformational change of Leu¹⁰⁴-His¹⁰⁵ result in hydrophobic pockets connect with the active center of ferrous ion.

2) In both the substrate/product binding states, we find that C76 is closer to Leu104 and His105, so C76 in channel II was selected for mutation (Fig. 4), the point mutations of C76Q or C76E completely abolished the enzymatic activity of NicX; in addition, C76A retained the capacity of transforming DHP into NFM. We speculate that the bulky residues (Glutamine or Glutamic acid) take up more space than Cysteine, and affect the range of movement of L104 and H105, potentially disrupting the ability to guide and stabilize the substrate to the appropriate position to initiate the reaction. Our data for the activity of the NicX variants support the review's view

about the function of H105 is in orienting the substrate. This is shown in the results section (Page 11 line 215-231), and is discussed in detail in the discussion of the revised manuscript.

Figure 4. A conformational change of L104 and H105. *A*, In NFM bound subunits, Cys⁷⁶ is close to Leu¹⁰⁴ at a distance of 4.9 Å, away from His¹⁰⁵ at a distance of 7.7 Å. *B*. In DHP bound subunits, Cys⁷⁶ is close to His¹⁰⁵ at a distance of 4.1 Å, away from Leu¹⁰⁴ at a distance of 7.7 Å.

(10) L172. Table S1 should be referred to at the end of the sentence.

Response: Thank you for the great suggestion. Corrected.

➤ **Solution:** Table S1 was added as suggested.

(11) L174. Table S2 should be referred to at the end of the sentence.

Response: Thank you for the great suggestion. Corrected.

➤ **Solution:** We added the corresponding Table S1 as suggested.

(12) L180-181. The distances reported are different from the values in Fig. 4A.

Response: Thank you for the great suggestion. Corrected.

➤ **Solution:** We rechecked the data and corrected it.

(13) L190. In writing about the kinetic parameters of the variants, the authors incorrectly use “activity” throughout when they mean Kcat or turnover.

Response: Thank you for the great suggestion.

➤ **Solution:** Throughout the revised manuscript, we now write the correct activity phrase in place of the previous incorrect phrase.

(14) L196. Fig. 4B should be referred to at the end of the sentence.

Response: Thank you for the great suggestion. We added the corresponding Fig. 3A at the end of the sentence.

(15) L199. Table 3 should be Table S2.

Response: Thank you for the great suggestion. Corrected.

➤ **Solution:** Table S2 was added.

(16) L199-201. There is a mixing of oxidation states in this sentence in which the first half of the sentence refers to binding ferric iron (which is likely the correct oxidation state in the ICP-MS) so it does not directly prove the ability of the R293A variant to bind ferrous iron. In addition, Asp is a perfect ligand for iron without Arg, and so the role of Arg may be primarily steric to position the Asp correctly for ligation (although this does not preclude the possibility of an electrostatic role promoting the chemical mechanism).

Response: Thank you for the professional suggestion. Owing to our previously incorrect diction, our intended meaning with this sentence was obscured. In fact, the FeCl₂ compound was used in these experiments, and we were trying to state is that Arg³²⁰ functions in maintaining the correct conformation of Asp³²⁰. We have reworded this sentence as suggested by the reviewer. See in Page 9 line185-188.

“Pursuing this, we conducted ICP-MS assays and found that the NicX^{R293A} variant lost the ability to bind a ferrous ion (Table S2). This result indicates that the role of Arg²⁹³ may be primarily steric, apparently functioning to position the Asp³²⁰ correctly for ligation.”

➤ **Solution:** We revised the sentence according to the suggestions of the reviewer.

(17) L204. Fig. 3 should be referred to at the end of the sentence.

Response: Thank you for the great suggestion. Corrected.

➤ **Solution:** The corresponding Figures (Fig. 4A ,4B) added at the end of the sentence.

(18) L205. Add “in the DHP bound subunits” to the end of the sentence.

Response: Thank you for the great suggestion. We have rearranged the content of this paragraph, and this particular sentence has been removed from the revised manuscript.

(19) L207. The distance is labeled as 3.0 angstroms in Fig. 4C.

Response: Thank you for the great suggestion. Corrected.

➤ **Solution:** We rechecked the data and corrected it.

(20) L218. Reference 19 is not the correct reference as there is nothing in that paper regarding the role of Glu248. I think they mean Liu, S., Su, T., Zhang, C., Zhang, W.M., Zhu, D., Su, J., Wei, T., Wang, K., Huang, Y., Guo, L., Xu, S., Zhou, N.Y., Gu, L. (2015) J Biol Chem 290 24547-24560.

Response: Thank you for the great suggestion. We reversed the order of the references. In the revised manuscript, we have carefully checked the serial numbers of the references.

(21) L227. “(2.4 and 2.6 Å)” only references the distances from the two carboxylate O of E177. I believe it should be “(His, 2.8 angstroms; Glu, 2.4 and 2.6 angstroms)”.

Response: Thank you for the great suggestion. We have revised the sentence and corrected it.

(22) L240. Fig. 6 is referenced before Fig. 5 in the text. Reverse the numbering.

Response: Thank you for the great suggestion. The figures have been redrawn, and we rearranged the order of the panels.

(23) Include a figure with 2 panels that clearly shows all the active site side-chains whose roles were probed by mutagenesis, along with the metal for both a DHP and NFM bound subunit from the same orientation long with a third panel of the overlay. These can be in stereo if it is too crowded. I think such a figure really needs to be in the main manuscript.

Response: Thank you for the great suggestion. We reworked the figure as suggested by the reviewer.

➤ **Solution:** We revised the figure as the followings.

Figure 3. Active site of NicX. *A*, Active site of NicX with bound DHP. The carbon atoms of DHP are in yellow. Green, red, blue and purple represent carbon, oxygen, nitrogen and Fe atoms respectively. *B*, Active site of NicX with bound NFM. The

carbon atoms of NFM are in yellow. Cyan, red, blue and purple represent carbon, oxygen, nitrogen and Fe atoms respectively. C, A stereoview of the DHP bound subunit (green) and NFM bound subunit (cyan), the superimposition was done on the whole subunit. Iron and solvent molecules are shown as purple and red spheres, respectively. Distances are given in angstroms.

(24) L268. “ferric” should be “ferrous”.

Response: Thank you for the great suggestion. Corrected.

➤ **Solution:** “ferric” replaced by “ferrous”.

(25) L293. “was disappeared, thereby” I think should be “are lost causing”.

Response: Thank you for the great suggestion. We completely rewrote the discussion section, and this sentence was removed in revision version.

(26) L326. Reference Table S1 at the end of this sentence.

Response: Thank you for the great suggestion. We rewrote the discussion section, and the content of comparisons with PnpCD have been removed, so this sentence is not present in the revised manuscript.

(27) L384. What was the ferric compound used in the ICP-MS assay?

Response: FeCl₂ was used in the ICP-MS assay.

(28) L397. “cryofreezing” should be “cryocooling”.

Response: Thank you for the great suggestion. Corrected.

➤ **Solution:** The “cryofreezing” was replaced by “cryocooling”.

(29) Table 1. The B-factors for the two structural models are quite high, but are reflective of the overall Wilson B-factors for the two structures. Include the Wilson B-factors in Table 1.

Response: Thank you for the great suggestion. We added the Wilson B-factors in Table 1 as suggested by the reviewer.

TABLE 1 Data collection and refinement statistics

	SeMet-NicX	NicX	NicX in complex with DHP and NFM
Data collection			
Space group	P2₁2₁2	P2₁2₁2	P2₁2₁2
Wavelength (Å)	0.97918	0.97892	0.97918
Cell dimensions			
a, b, c (Å)	125.92, 144.13, 118.89	125.961, 143.75, 118.69	126.67, 145.51, 118.98
α , β , γ (°)	90.00, 90.00, 90.00	90.00, 90.00, 90.00	90.00, 90.00, 90.00
Number of molecules/asymmetric unit	6	6	6
Resolution range (Å) (outer shell)	50-2.28 (2.40-2.28)	50.00-2.00 (2.03-2.00)	50-2.20 (2.24-2.20)
Completeness (%) (outer shell)	99.2 (99.6)	100.0 (97.3)	99.8 (99.2)
Redundancy (outer shell)	6.0 (6.2)	12.3 (12.3)	12.8 (13.1)
Total observations	585113	1763979	1453302
Unique reflections	98134	142933	113279
Wilson B factor (Å ²)	50.13	22.91	27.51
R _{merge} (%) (outer shell)	9.1 (65.1)	10.1 (95.3)	10.6 (95.9)
I/ σ _I (outer shell)	8.7 (3.0)	25 (2.67)	24.4 (3.7)
Refinement			
Resolution range (Å)	30-2.28	47.41-2.00	27.59-2.20
R _{work} /R _{free} (%)	21.1/26.7	17.12/21.33	18.3/23.8
Average B-factors(Å ²)			
Protein, Metal ion, Water, Substrate/Product	67.3, 97.2, 55.1, -	27.75, 38.04, 40.28, -	29.52, 45.6, 38.8, 42.9/55.1
Root mean square deviations			
Bond angles (°), Bond lengths (Å)	0.014, 1.644	0.010, 1.625	0.008, 0.908
Number of atoms			
Protein/Substrate/Water	16371/0/241	16334/0/1514	16576/52/940
Ramachandran plot			
Most favored, Allowed, Disallowed (%)	95.1, 4.7, 0.1	95.81, 3.66, 0.53	95.38, 4.23, 0.38

(30) Fig. 2A. I can only see 5 chain colors in the figure. Make the 6th subunit color more distinct so it can easily be seen.

Response: Thank you for the great suggestion. We reworked Fig. 2A as suggested by the reviewer.

➤ **Solution:** We revised the figure as the following.

Figure 2. Structure of NicX. *A*, Overall structure of NicX. Ribbon plot representation of the NicX hexamer. *B*, overall structure of a protomer of NicX. The α -helices, β -sheets and loops are in red, yellow and green respectively. Secondary structure elements of NicX are labelled. *C*, coordination sites of NicX-Fe(II). The $2F_o-F_C$ electron density map is contoured at 1σ ; water 1 molecule is opposite Asp³²⁰, while water 2 molecule is opposite Ser³⁰². *D*, the 14 Å-deep substrate-binding pocket on the NicX surface, two channels are observed. *E*, Two tunnels are separated by residues His¹⁰⁵ and Glu³⁰⁸, observed in substrate-free structure or NFM complex structure. *F*, In the DHP bound

subunits, channel II was blocked by residues Leu¹⁰⁴ and His¹⁰⁵.

(31) Fig. 3. Panels C and D are not informative. Replace these with panels suggested in point (23).

Response: Thank you for the great suggestion. The figure was revised.

➤ **Solution:** The figure was revised as the following.

Figure 3. Active site of NicX. *A*, Active site of NicX with bound DHP. The carbon atoms of DHP are in yellow. Green, red, blue and purple represent carbon, oxygen, nitrogen and Fe atoms respectively. *B*. Active site of NicX with bound NFM. The carbon atoms of NFM are in yellow. Cyan, red, blue and purple represent carbon, oxygen, nitrogen and Fe atoms respectively. *C*, A stereoview of the DHP bound subunit (green) and NFM bound subunit (cyan), the superimposition was done on the whole subunit. Iron and solvent molecules are shown as purple and red spheres, respectively. Distances are given in angstroms.

(32) L590-91. What is the grey mesh in the video?

Response: The grey mesh presents the electron cloud of the substrate.

We would again like to take this opportunity to thank the reviewer for the exceedingly helpful and insightful comments about how to improve our study and dramatically improve the quality and scientific purport of our interpretations and our discussion. Thank you very much!

REVIEWER COMMENTS

Reviewer #1 (Remarks to the Author):

The authors have responded to my points about the proposed mechanism, and have cited some relevant papers, although there are others dealing specifically with acid-base chemistry in the extradiol catechol dioxygenase mechanism.

In Figure 5, they have proposed three possible catalytic mechanisms. There are some points and mistakes regarding these possibilities:

- pathway A involves an epoxide intermediate, which has been suggested as a possible intermediate in the extradiol catechol deoxygenates, although does not match most of the experimental data available. The authors should make sure that they include oxidation states on iron for each intermediate.
- pathway B is possible, although the authors should mention that two one-electron transfers are needed to form the 2nd peroxide intermediate, and the oxidation states for the 2nd and 3rd intermediates should be +2 (2nd not labelled, 3rd labelled as +3)
- pathway C is a variation on pathway B, but with dioxygenase binding in a different co-ordination site. There are several mistakes in the 2nd and 3rd intermediates drawn. If the authors mean that two 1-electron transfers take place on the 1st intermediate, then that would form a semiquinone and iron (II)-superoxide (not bonded together); C-O bond formation would then form a hydroperoxide related to the 2nd structure, but with a carbonyl group on the substrate (not a radical), and 2+ on iron (not 3+); Criegee rearrangement of this would then form what is the 4th intermediate shown, which is drawn correctly.

In the text that discusses the mechanism (p 12 lines 250-252), "semiquinone peroxide" is incorrect. As noted above, a semiquinone is formed before C-O bond formation; after C-O bond formation there is a hydroperoxide, but it does not contain a radical.

The authors have also discussed a possible proton donor. An Asp proton donor seems very unlikely to me, because Asp would be deprotonated at pH 7, and could not therefore act as a proton donor. Arg-293 could possibly be a proton donor, although the pKa of arginine residues is normally too high to act as a catalytic proton donor, however there are some examples in the literature where Arg residues have been proposed to act as proton donors (maybe with lower pKa values), the authors might like to search for examples of these in the literature.

Reviewer #2 (Remarks to the Author):

This revised version of the study of the structure and mechanism of NicX is considerably improved both in readability and separation of results and hypothesis. The first structure of an Fe(II) dioxygenase that can open a pyridine ring, which has a Ser ligand necessary for Fe(II) binding, (and function?), and which binds its substrate away from the iron is a significant advance. Extensive use of mutagenesis is used to establish the iron ligation and probe the roles of several residues in the second sphere. These results and computations support the idea that the reaction of O₂ with the substrate bound away from the iron is possible. Moreover, two alternative mechanisms are presented as possible with the reaction of the remotely bound substrate favored in arguments and computations presented in the discussion. This discussion, while speculative seems justified based on the structural data.

From my reading of the computational results, all of the pathways are ultimately downhill energetically and Pathway 1A and II cannot be distinguished based on energetics alone. If this reading is correct, that should be stated in the text. Also, from the figures shown, it appears that the substrate is properly oriented for Pathway 1A, albeit with a 4 Å translocation. It is not obvious that Pathway II can occur without a rotation of the substrate, but may only be the perspective of the figure. If the bridging peroxo can form without a shift of the substrate, this should be stated in

the text. The figure showing the intermediate treated in the computation indicates the starting point is the resting enzyme with waters bound to the Fe(II). The problem I noted before with Pathway II is that O₂ has a rather poor affinity for non-heme Fe(II). Consequently computations of extradiol dioxygenases in the past have shown little reaction in the absence of another source of an electron beyond the Fe(II). In the extradiols, this is the Fe-bound substrate, whereas in the Rieske dioxygenases, it is the Rieske Fe-S cluster. It is not clear why this should be different in the current case, especially for Pathway II, which shows a considerably favorable reaction for some multiplicities. This is an important part of the mechanism, so if there is insight into why this enzyme is different, it would be welcome. Addition of substrate to the Fe(II) ligation in pathway 1A should address this problem directly, so it is mysterious why the energetics are not even more favorable. One way to address this problem is to assume that the O₂ binding in Pathway II is weak and reversible. It is stabilized by the electron transfer from the substrate as the bound superoxo species attacks. This idea also does not seem to correlate with the computation, which shows high affinity for O₂ before attack in Pathway II. Beyond these comments, I think this is an excellent contribution.

Line 31 – spelling ---- pyridine

Line 51 – poor transition in the sentence between Fe ligands and active site residues.

Line 77 – catechol intradiol dioxygenases are not Fe(II), they are Fe(III).

Line 95 – if the Ser ligand is demonstrated in this work, the verb should be “is” not “was”

Line 228 - Leu

I think the authors have done a good job of responding to the comments, at least for reviewers 2 and 3. Questions remain concerning the mechanism proposed by the authors, which was questioned by both of these reviewers. However, the mutagenesis and the computations conducted are supportive and the authors now more clearly state that there is more work to be done to verify the proposal. They have attempted to more critically evaluate their proposed mechanism vs the likely alternative analogous to the extradiol mechanism. I think the novel structure of this subclass and the evidence in support of a new type of ring cleaving dioxygenase mechanism is a good step forward and will be of broad interest, especially among bio-organic and bio-inorganic chemists and biochemists.

A few minor things:

Line 77 – The intradiol catecholic dioxygenases are Fe(III) not Fe(II). Gentisate dioxygenases would be another class. Since the list is not inclusive, perhaps use “among them” instead of “including”

Line 79-80 - 2-Hydroxyethylphosphonate dioxygenase does not use a Pterin

Line 95 – How was Ser identified as a ligand without a structure? Add a reference

Line 160 – soaked not socked

Reviewers' comments:

Response to Reviewer 1

REVIEWER #1

1) The authors have responded to my points about the proposed mechanism, and have cited some relevant papers, although there are others dealing specifically with acid-base chemistry in the extradiol catechol dioxygenase mechanism. In Figure 5, they have proposed three possible catalytic mechanisms. There are some points and mistakes regarding these possibilities:

Response: We are again grateful to the reviewer for the helpful guidance about how to improve our study and manuscript. We respond to comments in a point-by-point manner, below.

2) - pathway A involves an epoxide intermediate, which has been suggested as a possible intermediate in the extradiol catechol deoxygenates, although does not match most of the experimental data available. The authors should make sure that they include oxidation states on iron for each intermediate.

Response: Thank you for focusing our attention on this important issue. We have solved this problem with Figure 5; we have now revised Figure 5 and now include the oxidation states on iron for each intermediate.

Figure 5. Plausible pathways for NicX-catalyzed DHP degradation, in which dioxygen attacks either from the equatorial position (A and B), or the apical position (C). (Pathway IA and IB denote that the nitrogen and oxygen atoms of DHP bind the metal ferrous center, respectively. The C6 is the most vulnerable position in DHP, with an f^+ value of 0.148 based on Fukui function analysis).

3) - pathway B is possible, although the authors should mention that two one-electron transfers are needed to form the 2nd peroxide intermediate, and the oxidation states for the 2nd and 3rd intermediates should be +2 (2nd not labelled, 3rd labelled as +3).

Response: Thanks for the reviewer’s helpful suggestions. We have corrected the oxidation states on the irons and the revised text now reads: “Subsequently, two one-electron transfers are needed to form the peroxide intermediate, which can be followed by a Criegee rearrangement to yield a 7-membered-ring lactone and a ring-opened product; either of these scenarios would result in a substrate-bound iron

arrangement similar to the classic extradiol catechol dioxygenases (Pathway IA & IB)” (See Page 12 line 243-247).

4) - pathway C is a variation on pathway B, but with dioxygenase binding in a different co-ordination site. There are several mistakes in the 2nd and 3rd intermediates drawn. If the authors mean that two 1-electron transfers take place on the 1st intermediate, then that would form a semiquinone and iron (II)-superoxide (not bonded together); C-O bond formation would then form a hydroperoxide related to the 2nd structure, but with a carbonyl group on the substrate (not a radical), and 2+ on iron (not 3+); Criegee rearrangement of this would then form what is the 4th intermediate shown, which is drawn correctly.

Response: Thanks for the reviewer’s helpful suggestions. We have redrawn this pathway and corrected the oxidation states on the irons (See Figure 5). The first one-electron transfer would occur between the DHP and the adjacent dioxygen molecule at the apical position directly. The lowest-energy reacting complex (the first structure) was calculated to be either quintet or septet, with a very small energy gap of 0.3 kcal/mol. We did not locate the one-electron transfer radical species in square brackets yet, which could be a non-covalent bound complex of semiquinone and iron(II)-superoxide.

Figure 5. Plausible pathways for NicX-catalyzed DHP degradation, in which dioxygen attacks either from the equatorial position (A and B), or the apical position (C). (Pathway IA and IB denote that the nitrogen and oxygen atoms of DHP bind the metal ferrous center, respectively. The C6 is the most vulnerable position in DHP, with an f^+ value of 0.148 based on Fukui function analysis).

5) In the text that discusses the mechanism (p 12 lines 250-252), "semiquinone peroxide" is incorrect. As noted above, a semiquinone is formed before C-O bond formation; after C-O bond formation there is a hydroperoxide, but it does not contain a radical.

Response: Thanks for the reviewer's helpful suggestions. We have corrected "the semiquinone peroxide intermediate" to "the peroxide intermediate" and "the biradical intermediate" to "the low-lying septet state".

6) The authors have also discussed a possible proton donor. An Asp proton donor

seems very unlikely to me, because Asp would be deprotonated at pH 7, and could not therefore act as a proton donor. Arg-293 could possibly be a proton donor, although the pKa of arginine residues is normally too high to act as a catalytic proton donor, however there are some examples in the literature where Arg residues have been proposed to act as proton donors (maybe with lower pKa values), the authors might like to search for examples of these in the literature.

Response: Thanks for the reviewer's helpful suggestions. Given that many possible catalytic proton donors have been demonstrated for O-O cleavage, we have rewritten Lines 332-334 to "Third, the consequent O-O cleavage could be promoted by neighboring proton donor candidates such as imidazolium of His¹⁰⁵, neutral form of Asp³²⁰, aromatic N-H of DHP, and even guanidinium of Arg²⁹³ (39, 40)."

References

[39] Keenholtz, R. A., Mouw, K. W., Boocock, M. R., Li, N. S., Piccirilli, J. A. & Rice, P. A. Arginine as a general acid catalyst in serine recombinase-mediated DNA cleavage. *J. Biol. Chem.* **288**, 29206–29214 (2013).

[40] Burke, J. R., La Clair, J. J., Philippe, R. N. & Noel, J. P. Bifunctional substrate activation via an arginine residue drives catalysis in chalcone isomerases. *ACS Catal.* **9**, 8388–8396 (2019).

Response to Reviewer 2

REVIEWER #2

1) This revised version of the study of the structure and mechanism of NicX is considerably improved both in readability and separation of results and hypothesis. The first structure of an Fe(II) dioxygenase that can open a pyridine ring, which has a Ser ligand necessary for Fe(II) binding, (and function?), and which binds its substrate away from the iron is a significant advance. Extensive use of mutagenesis is used to establish the iron ligation and probe the roles of several residues in the second sphere. These results and computations support the idea that the reaction of O₂ with the substrate bound away from the iron is possible. Moreover, two alternative mechanisms are presented as possible with the reaction of the remotely bound substrate favored in arguments and computations presented in the discussion. This discussion, while speculative seems justified based on the structural data.

Response: Thank you for your continued time and effort in reviewing our study. Your support and recognition of progress is a great encouragement to us.

2) From my reading of the computational results, all of the pathways are ultimately downhill energetically and Pathway 1A and II cannot be distinguished based on energetics alone. If this reading is correct, that should be stated in the text.

Response: Thanks for the reviewer's helpful suggestions. This is now clearly stated in the revised version. See Page 17 line 327-329. "We consider the different catalytic pathways of NicX, wherein the dioxygen occupies the equatorial position (Pathway I)

or apical position (Pathway II), and these pathways are ultimately downhill energetically, they cannot be distinguished based on energetics alone.”. Please see

Figure 5.

Figure 5. Plausible pathways for NicX-catalyzed DHP degradation, in which dioxygen attacks either from the equatorial position (A and B), or the apical position (C). (Pathway IA and IB denote that the nitrogen and oxygen atoms of DHP bind the metal ferrous center, respectively. The C6 is the most vulnerable position in DHP, with an f^+ value of 0.148 based on Fukui function analysis).

3) Also, from the figures shown, it appears that the substrate is properly oriented for Pathway 1A, albeit with a 4 Å translocation. It is not obvious that Pathway II can occur without a rotation of the substrate, but may only be the perspective of the figure. If the bridging peroxo can form without a shift of the substrate, this should be stated in the text.

Response: Thanks for the reviewer’s helpful suggestions. This is now clearly stated in the revised version. See Page 15 line 326-329. “Alternatively, in Pathway II the

bridging peroxo can undergo a ~ 0.5 Å shift of the substrate and a $\sim 20^\circ$ rotation of the substrate, a scenario that would retain the intermediate in the hydrogen bond network.”.

4) The figure showing the intermediate treated in the computation indicates the starting point is the resting enzyme with waters bound to the Fe(II). The problem I noted before with Pathway II is that O₂ has a rather poor affinity for non-heme Fe(II). Consequently computations of extradiol dioxygenases in the past have shown little reaction in the absence of another source of an electron beyond the Fe(II). In the extradiols, this is the Fe-bound substrate, whereas in the Rieske dioxygenases, it is the Rieske Fe-S cluster. It is not clear why this should be different in the current case, especially for Pathway II, which shows a considerably favorable reaction for some multiplicities. This is an important part of the mechanism, so if there is insight into why this enzyme is different, it would be welcome.

Response: Thanks for the reviewer’s helpful suggestions. Based on our understanding, in heme enzyme, oxygen molecules are located between the substrate and iron, and the substrate can directly transfer electrons to oxygen. In NicX, the substrate DHP can directly transfer electrons to oxygen by current calculations and structural results. The arrangement of Fe-oxygen-substrate in NicX is similar to that of the heme enzyme, this is why NicX is different from Extradiol dioxygenase or Rieske dioxygenases. The apical arrangement would be similar to a P450-like arrangement, and Ser³⁰² could exert a cysteine-like catalytic role (Pathway II). Such a

substrate-iron indirect arrangement was the first time proposed in non-heme iron enzyme, however, it is popular in heme-Fe enzyme.

5) Addition of substrate to the Fe(II) ligation in pathway 1A should address this problem directly, so it is mysterious why the energetics are not even more favorable. One way to address this problem is to assume that the O₂ binding in Pathway II is weak and reversible. It is stabilized by the electron transfer from the substrate as the bound superoxo species attacks. This idea also does not seem to correlate with the computation, which shows high affinity for O₂ before attack in Pathway II. Beyond these comments, I think this is an excellent contribution.

Response: Thanks for the reviewer's helpful suggestions. We presently have reaction energetics data and have examined all three pathways in a step-by-step manner. The binding affinity of dioxygen looks similar at both the equatorial and apical positions, indicating that first-step dioxygen loading is apparently reversible between the two positions. It is thus possible, perhaps even probably, that the reaction barrier for the following electron-transfer from the substrate is critical for the enzyme. We are continuing our computational investigations but these complex questions remain unanswered at present. Please see Figure 5.

Figure 5. Plausible pathways for NicX-catalyzed DHP degradation, in which dioxygen attacks either from the equatorial position (A and B), or the apical position (C). (Pathway IA and IB denote that the nitrogen and oxygen atoms of DHP bind the metal ferrous center, respectively. The C6 is the most vulnerable position in DHP, with an f^+ value of 0.148 based on Fukui function analysis).

6) Line 31 - spelling ---- pyridine

Response: Thank you for this reminder; we have corrected this in the revised manuscript.

Solution: “2,5-Dihydroxypridine” was replaced by “2,5-Dihydroxypyridine”

7) Line 51 - poor transition in the sentence between Fe ligands and active site residues.

Response: Thank you for this reminder; we have revised this sentence for clarity.

Solution: We reworded the sentence, see Page 3 line 47–52. “We demonstrate that

NicX employs a rare four-residue (His²⁶⁵, Ser³⁰², His³¹⁸, and Asp³²⁰) coordination for Fe (II), and uses Leu¹⁰⁴-His¹⁰⁵ as active-site residues for guiding and stabilizing the substrate (DHP) to the appropriate position to initiate the ring-opening reaction to produce the product *N*-formylmaleamic acid (NFM)”.

8) Line 77 - catechol intradiol dioxygenases are not Fe(II), they are Fe(III).

Response: Thank you for this reminder; we have corrected this in the revised manuscript.

Solution: Thanks for your helpful suggestions, this sentence has been reworded in the revised manuscript. See page5 line73–78 “These enzymes can be classified into several different groups based on their structural characteristics, reactivity, and specific requirements for catalysis, among them: (I) Extradiol cleaving catechol dioxygenases, (II) Rieske oxygenases, (III) Alpha-ketoglutarate dependent enzymes, (IV) Cysteine dioxygenases, and (V) Pterin-dependent hydroxylases (13, 14)”.

9) Line 95 - if the Ser ligand is demonstrated in this work, the verb should be “is” not “was”

Response: Thank you for this reminder; we have corrected this in the revised manuscript.

Solution: This sentence has been reworded in the revised manuscript. See page 5 line 94–97 “Interestingly, NicX Ser³⁰² coordinates the iron(II) ion; a similar metal ion-interacting serine residue has been reported for a dialkylglycine decarboxylase,

Cu⁺-ATPases and for transcriptional activators like CueR and GolS (25–28)”

10) Line 228 - Leu

Response: Thank you for this reminder; we have corrected this in the revised manuscript.

Solution: “Lue” was replaced by “Leu”

I think the authors have done a good job of responding to the comments, at least for reviewers 2 and 3. Questions remain concerning the mechanism proposed by the authors, which was questioned by both of these reviewers. However, the mutagenesis and the computations conducted are supportive and the authors now more clearly state that there is more work to be done to verify the proposal. They have attempted to more critically evaluate their proposed mechanism vs the likely alternative analogous to the extradiol mechanism. It think the novel structure of this subclass and the evidence in support of a new type of ring cleaving dioxygenase mechanism is a good step forward and will be of broad interest, especially among bio-organic and bio-inorganic chemists and biochemists.

Response: Thank you for your time and effort in reviewing our manuscript. We appreciate your help about how to improve our study.

A few minor things:

1) Line 77 - The intradiol catecholic dioxygenases are Fe(III) not Fe(II). Gentisate dioxygenases would be another class. Since the list is not inclusive, perhaps use “among them” instead of “including” .

Response: Thanks for your helpful suggestions, this sentence has been reword in the revised manuscript. See page5 line73–78 “These enzymes can be classified into several different groups based on their structural characteristics, reactivity, and specific requirements for catalysis, among them: (I) Extradiol cleaving catechol dioxygenases, (II) Rieske oxygenases, (III) Alpha-ketoglutarate dependent enzymes,

(IV) Cysteine dioxygenases, and (V) Pterin-dependent hydroxylases (13, 14)”.

2) Line 79-80 - 2-Hydroxyethylphosphonate dioxygenase does not use a Pterin.

Response: Thanks for your the helpful pointer here. This part has been removed from the revised manuscript.

3) Line 95 - How was Ser identified as a ligand without a structure? Add a reference.

Response: Thank you very much for this important comment. It has been previously reported that serine was identified as a metal ligand without obtaining the structure^[1]: González-Guerrero, M. *et al.* (2008) used mutation experiments to determine the amino acid residues involved in metal ion binding. Previous studies have also indicated that serine is a rare ligand for non-heme Fe(II) dependent dioxygenases; however, it has been commonly reported as a ligand of other types of protein, such as transcriptional activators (*e.g.*, CueR and GolS)^[2,3]. This content has been reworded the new manuscript (see Page 5 line 94-97): “NicX Ser³⁰² coordinates the iron(II) ion; a similar metal ion-interacting serine residue has been reported-for a dialkylglycine decarboxylase, Cu⁺-ATPases and for transcriptional activators like CueR and GolS.”.

References

- [1] González-Guerrero, M., Eren, E., Rawat, S., Stemmler, T. L., & Argüello, J. M. (2008). Structure of the two transmembrane Cu⁺ transport sites of the Cu⁺-ATPases. *The Journal of biological chemistry*, 283(44), 29753–29759.
- [2] Ibáñez, M. M., Checa, S. K., & Soncini, F. C. (2015). A single serine residue determines selectivity to monovalent metal ions in metalloregulators of the MerR family. *Journal of bacteriology*, 197(9), 1606–1613.
- [3] Hobman J. L. (2007). MerR family transcription activators: similar designs, different specificities. *Molecular microbiology*, 63(5), 1275–1278.

4) Line 160 - soaked not socked.

Response: Thank you for this reminder; we have corrected this in the revised manuscript.

Solution: “socked” was replaced by “soaked”.

REVIEWERS' COMMENTS

Reviewer #1 (Remarks to the Author):

The authors have responded to the points raised. The proposed mechanisms are now more reasonable.

Reviewer #2 (Remarks to the Author):

The authors have addressed my concerns with this manuscript. I think it is a valuable contribution that introduced several new aspects of dioxygenase catalysis as described in the previous reviews.